# Population reconstructions for humans and megafauna suggest mixed causes for North American Pleistocene extinctions

Jack M. Broughton[1] & Elic M. Weitzel [2]

Dozens of large mammals such as mammoth and mastodon disappeared in North America at the end of the Pleistocene with climate change and "overkill" by human hunters the most widely-argued causes. However, the population dynamics of humans and megafauna preceding extinctions have received little attention even though such information may be telling as we expect increasing human populations to be correlated with megafaunal declines if hunting caused extinctions. No such trends are expected if climate change was the primary cause. We present tests of these hypotheses here by using summed calibrated radiocarbon date distributions to reconstruct population levels of megafauna and humans. The results suggest that the causes for extinctions varied across taxa and by region. In three cases, extinctions appear linked to hunting, while in five others they are consistent with the ecological effects of climate change and in a final case, both hunting and climate change appear responsible.

[1] Department of Anthropology and Archaeological Center, University of Utah, 260 Central Campus Drive, Suite 4444, Salt Lake City, UT 84112, USA.
[2] Department of Anthropology, University of Connecticut, 354 Mansfield Rd., Storrs, CT 06269, USA. These authors contributed equally: Jack M. Broughton, Elic M. Weitzel. Correspondence and requests for materials should be addressed to J.M.B. (email: jack.broughton@anthro.utah.edu)

Toward the end of the Pleistocene, North America lost 37 mammalian genera including over 70% of its megafauna, commonly defined as terrestrial taxa exceeding 44 kg[1]. Although part of a broader, global wave of late Quaternary extinctions[2–4] and general biotic upheaval[5–9], the North American losses were among the most severe[1]. The causes of these extinctions have been debated for centuries[10] with human hunting (i.e., "overkill") and the ecological effects of climate change the most widely cited principal drivers[1–11]. The subsequent impacts of megafaunal elimination on ecosystem features —from vegetation, fire, and small mammal dynamics to regional and global climate—are also increasingly recognized[7–9], as are the resulting implications for future landscape change should currently at-risk large mammals fall to extinction[7]. Although much effort has focused on the chronology of extinctions in relation to human arrival and changing climate[2–5,12–18], far less attention has focused on the population dynamics of humans and megafauna during the millennia preceding extinctions. Yet these data should be especially telling as to the relative roles of hunting and climate change in megafaunal extinctions. Fortunately, methods for reconstructing human and large mammal population histories have witnessed recent advances including approaches based on genetic diversity derived from ancient DNA[19–22] and those that involve the use of radiocarbon date frequencies as proxies for population levels[23–28].

We take the latter approach here based on the premise that temporal frequencies of radiocarbon dates can serve as population proxies since larger populations should produce and deposit greater amounts of dateable carbon[23–28]. This method has been increasingly refined and applied to investigate population histories in a wide range of contexts including the case of Pleistocene extinctions[14,28–30].

Although the potential role of human hunting in megafaunal extinctions has a deep history[10], modern formulations of the overkill hypothesis[31] explicitly link extinctions to the well-known Clovis archeological phenomenon. Dating to 13.15−12.85 ka, Clovis represents the earliest, widespread archeological culture in the New World and is readily identified by large, lanceolate, and fluted stone spear points known as Clovis points. In a handful of sites ($n = 15$)[32], Clovis points have been recovered in association with extinct fauna—the vast majority (80%) being mammoth (Mammuthus)— and it has long been assumed that Clovis represents the first big-game hunting groups in North America[12]. This position remains widely held despite continued discussion about the role that megafauna played in Clovis diets and growing evidence for a pre-Clovis human presence[12,33]. Critics of overkill have emphasized the small number of securely documented Clovis kill sites considering the magnitude of the slaughter implied[1,12] but advocates counter that the evidence is consistent with expectations given the narrow window of time over which the extinctions are believed to have occurred[31,34]. In any case, we expect that if Clovis hunting drove extinctions, megafaunal population declines should occur during Clovis times and be associated with significant human population increases. In other words, negative correlations should exist between human and megafaunal populations. This pattern is explicit in descriptions and model simulations of the overkill hypothesis[35,36], and is supported by foraging theory logic and the well-documented impacts of rising late Holocene human populations on surviving megafauna[37,38]. Overkill resulting from subsequent cultures (e.g., Folsom) would be suggested by negative correlations where significant megafaunal population declines leading to extinction occur after Clovis times.

No such trends are predicted if the ecological effects of climate change drove extinctions. In such cases, megafaunal declines should align with climatic and environmental conditions recognized to negatively impact large herbivores[39–41]. Climatic hypotheses for extinctions are varied and can be taxon-specific[3,21,30] but most relate to how changes in temperature, precipitation, and insolation seasonality resulted in habitat loss, simplification or fragmentation of plant communities, changes in plant nutrient quality and seasonal availability, or physiological stress[1,4,6,39,40].

Megafaunal extinctions resulting from these processes have been suggested for a range of late Pleistocene climatic episodes but recent work has suggested strong associations between extinction events and warm interstadials associated with Dansgaard-Oeschger events[2,3,9], such as the Bølling-Allerød (B-A; ~14.7−12.9 ka), as well as the unique conditions of the Younger Dryas (YD). For instance, using ancient DNA and radiocarbon records, Cooper et al.[2] derive detailed time series for multiple megafauna genetic clades or species in both Eurasia and Eastern Beringia and show that the period encompassing the B-A and YD is characterized by a higher frequency of extirpations or global extinctions than any other over the last 25,000 years. Since it is known that many now-extinct genera of North American megafauna survived the B-A[1,12–14], attention has focused on the potential role of conditions during the YD in driving extinctions in this region[5,12,15,16].

Most work suggests conditions of the YD resulted from a complex matrix of global climatic forcing mechanisms that may have been unique to the transition into the current inter-glacial cycle relative to earlier ones[42,43]. Although it is difficult to generalize the conditions of the YD across North America as the climate of the period clearly varied spatially, temporally, and seasonally[44], many YD paleoclimatic records suggest an abrupt return to glacial-like average annual temperatures, combined with peaks in insolation seasonality (i.e., cold winters and hot summers) and rapidly growing atmospheric $CO_2$[43–46]. Pronounced vegetational changes mark this period as well, but their extent and abruptness, again, exhibit substantial geographic variation[47–49]. This variability notwithstanding, the intersection of plant community reorganization with enhanced seasonality and shorter growing seasons is widely recognized to depress mammalian reproduction with magnified impacts on large-sized, long-lived, hypermorphic taxa[39–41,50]. Further, research suggests elevated $CO_2$ concentrations are associated with declines in plant nutrient (e.g., nitrogen) content and herbivore foraging efficiency[51,52], as well as increases in plant biochemical defenses[53].

Climate-based causes would thus be suggested in cases where megafaunal population declines that culminate in extinction occur during the B-A or YD and where human and megafaunal populations are positively correlated—rising and falling together —or uncorrelated. Finally, climatic causes would also be suggested insofar as megafaunal and human populations are uncorrelated and extinctions occur during other climatic episodes characterized by conditions potentially unfavorable to large herbivores (e.g., early Holocene).

To test these predictions concerning the dynamics of Pleistocene human and megafauna populations, we construct summed probability distributions (SPDs) of calibrated radiocarbon dates on a series of the best-dated megafauna taxa and anthropogenic materials and follow the methodology of Shennan et al.[54–57] to estimate the timing of population declines that culminate in extinction. Correlation analyses are used to assess the relationships between megafauna and human SPDs. The results suggest that the causes for megafauna extinctions varied across taxa and by region. In three cases, extinctions appear linked to Clovis hunting, while in five others they are consistent with the effects of YD climate change. In a final case, both human hunting and climate change appear responsible.

## Results

**Megafauna and human radiocarbon records**. We use the archeological radiocarbon record for the contiguous US derived from the Canadian Archaeological Radiocarbon Database (CARD)[58] for the period between 15.0 and 10.0 ka ($n = 938$ dates) and previously published direct dates on megafauna to model population trends. Representing the largest sample of its kind, we compiled 521 direct dates on megafauna specimens for the late Pleistocene from the contiguous United States (US) and southern Canada (Fig. 1; Supplementary Data 1). The sample includes 19 of the 37 extinct genera, or 22 total extinct taxa by including the relatively abundant date records for dire wolf (*Canis dirus*), Harrington's mountain goat (*Oreamnos harringtoni*), and American lion (*Panthera leo*). Despite this substantial sample, nearly half (48%) of the extinct Pleistocene mammalian genera are not represented by direct dates. In addition, just nine of the represented taxa—horse (*Equus*), camel (*Camelops*), dire wolf, Harrington's mountain goat, saber-toothed cat (*Smilodon fatalis*), mammoth, mastodon (*Mammut americanum*), Shasta ground sloth (*Nothrotheriops shastensis*), and giant bear (*Arctodus*)—make up 85% (443/521) of the total collection of dates. The sample dating between 20.0 and 10 ka ($n = 332$) is similar in composition with 18 extinct taxa represented and with the same nine comprising 93% (307/332) of the sample.

**Trends in the contiguous United States**. In the contiguous US, SPDs were generated separately for mammoth, mastodon, Shasta ground sloth, horse, and saber-toothed cat. At this broadest spatial scale, these taxa and humans show broadly similar patterns with population growth consistent with the null models occurring from ~15.0 ka through the warm and moist B-A interstadial until busts initiate between 13.0 and 11.9 ka (Fig. 2). For mammoth, horse, and saber-toothed cat, busts occur

during the latter half of the Clovis period. However, both Shasta ground sloth and mastodon do not exhibit significant population busts until after Clovis during the early YD at 12.65 ka, the same time that the human population first shows a significant decline. To explore the relationships between the megafaunal and human SPDs more directly, we calculated Spearman's rank correlation coefficients (Spearman's rho = $r_s$; two-tailed tests are used throughout) between them for the period of time between 15.0 and 11.7 ka. Again, significant negative correlations would be consistent with a human hunting impact and those are evident for mammoth ($r_s = -0.59$, $P < 0.0001$), horse ($r_s = -0.43$, $P < 0.0001$), and saber-toothed cat ($r_s = -0.66$, $P < 0.0001$) and technically for mastodon, though despite the significant $P$ value, the exceedingly low rho statistic indicates no meaningful relationship between these SPDs ($r_s = -0.04$, $P = 0.01$). In contrast, the relationship between the human and sloth SPDs is positive and highly significant ($r_s = 0.35$, $P < 0.0001$), suggesting those populations fluctuated together from extrinsic factors.

**Trends in the Southwest**. For the Southwest region, sloth populations bust during the early YD at 12.7 ka shortly before a human population decline occurs at 12.5 ka—the relationship between the human and sloth SPDs is positive and highly significant ($r_s = 0.66$, $P < 0.0001$; Fig. 3). Mammoths in this region, on the other hand, exhibit a more complex pattern. They exhibit an initial ~200-year post-Clovis bust beginning at 12.9 ka, followed by a return to trend for about the same duration, and then a second brief (~100-year) bust at 12.5, that is again followed by a ~400-year recovery. The final bust for mammoths leading to regional extinction occurs toward the end of the YD at 12.0 ka. The relationship between the human and mammoth SPDs is negative and significant ($r_s = -0.78$, $P < 0.0001$).

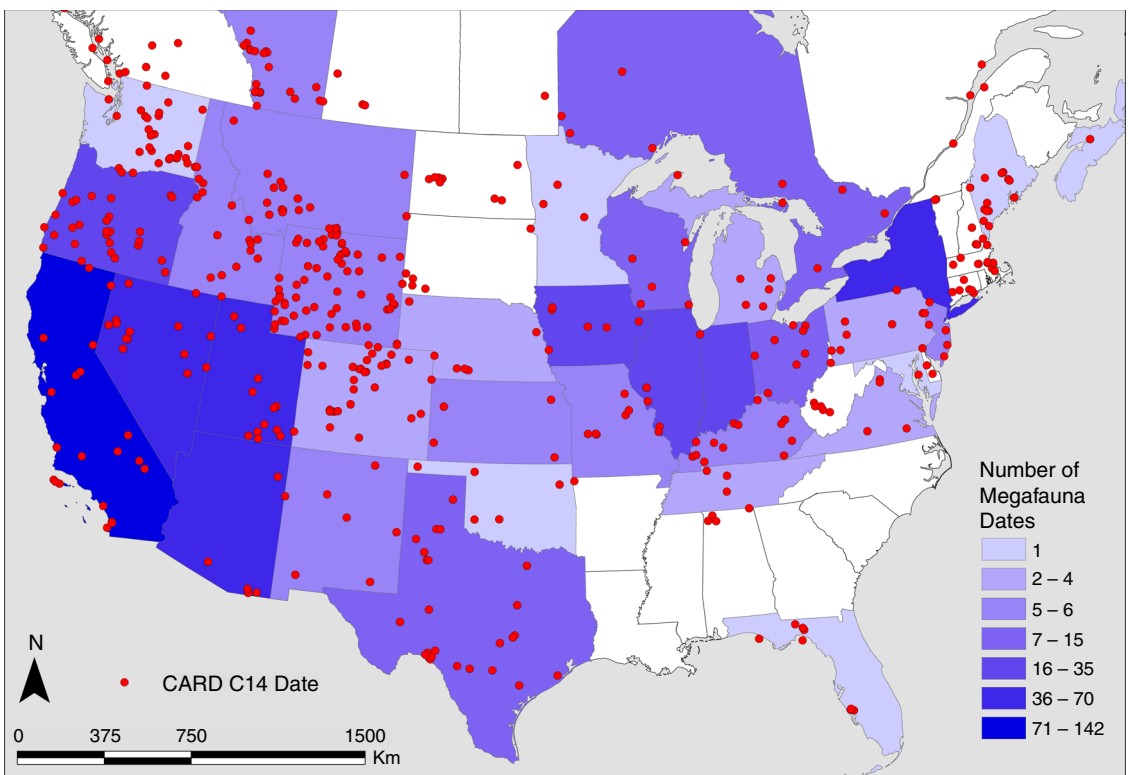

**Fig. 1** Map of the contiguous United States and southern Canada. Indicated are the frequency of dates by state or province for the directly dated megafauna sample and the distribution of archeological radiocarbon dates from the CARD used in this study. State and province imagery courtesy of ESRI[73]

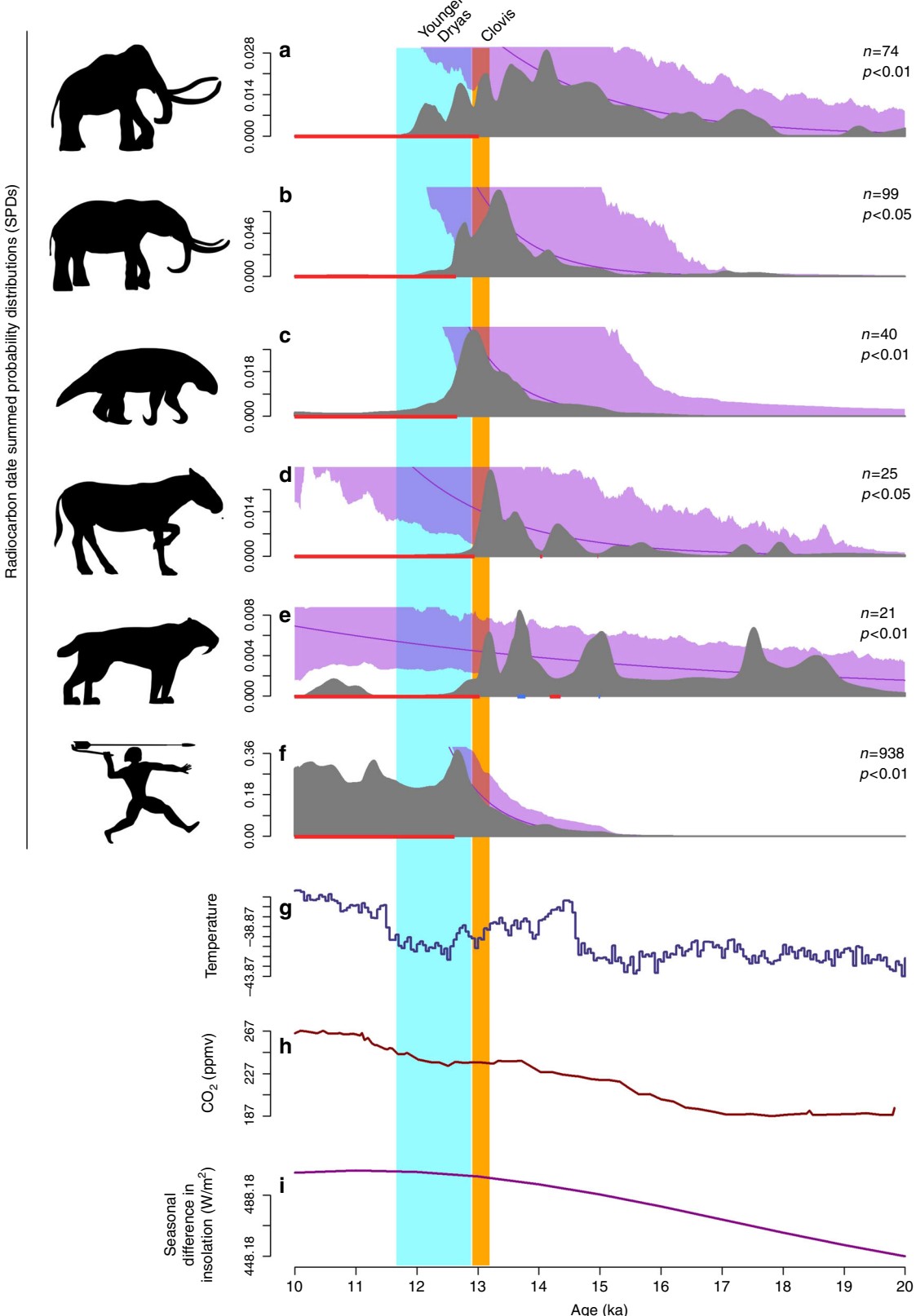

**Fig. 2** Megafaunal and human SPDs through time for the contiguous United States. Mammoth (**a**), mastodon (**b**), Shasta ground sloth (**c**), horse (**d**), saber-toothed cat (**e**), and human SPDs (**f**) are indicated in gray. The null model of exponential growth is indicated by the purple line; the purple shading denotes the 95% confidence interval around the null model (see Methods). Statistically significant deviations from the null model between 15 and 10 ka are indicated by the red (busts) and blue (booms) rugs at the base of each panel. Time series of $\delta^{18}O$ values from the NGRIP ice core (**g**), $CO_2$ from the Dome C, Antarctica, ice core (**h**), and insolation seasonality (**i**) are provided for comparison. Data sources are in Methods

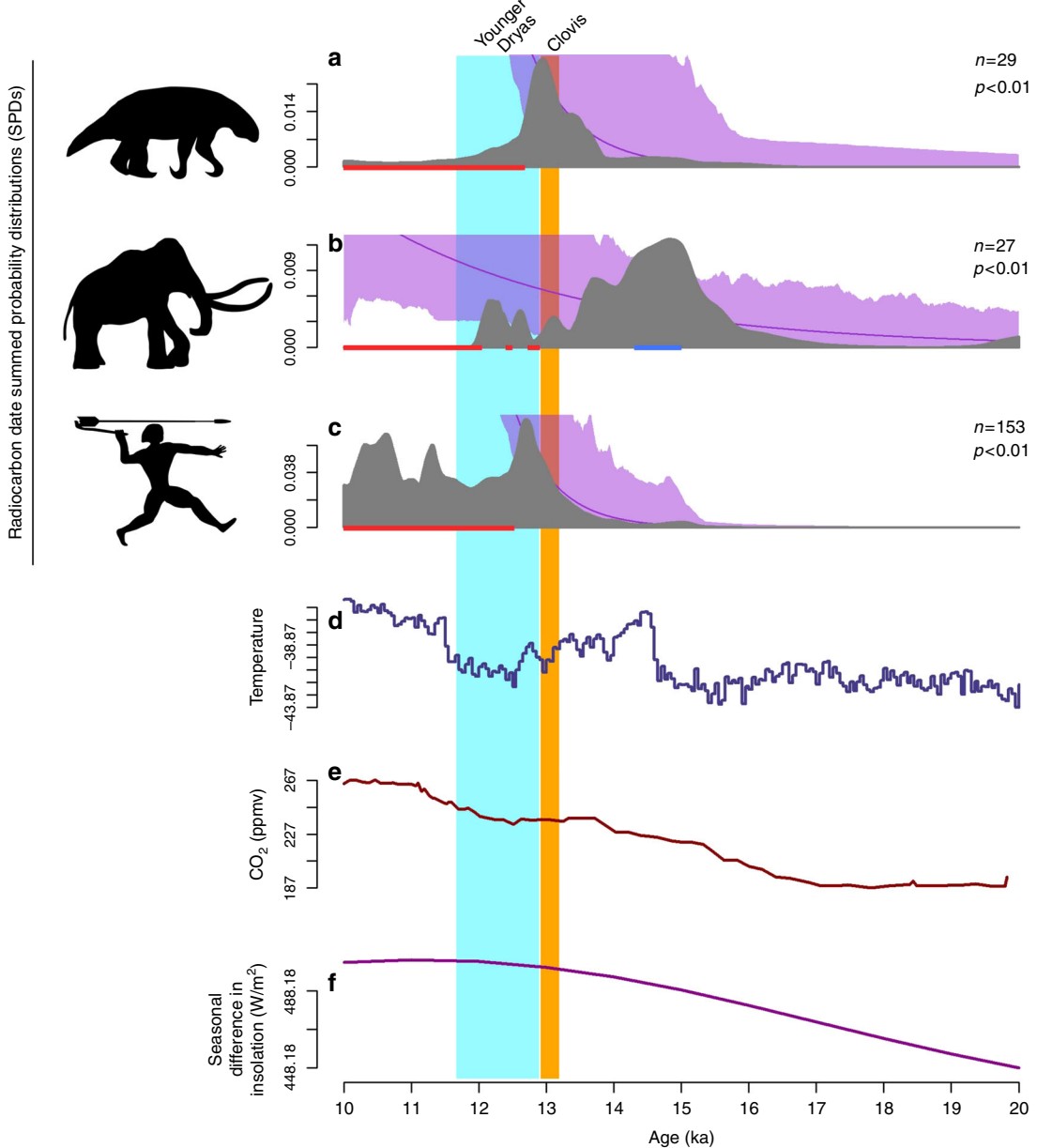

**Fig. 3** Megafaunal and human SPDs through time for the Southwest. Shasta ground sloth (**a**), mammoth (**b**), and human SPDs (**c**) are indicated in gray. The null model of exponential growth is indicated by the purple line; the purple shading denotes the 95% confidence interval around the null model (see Methods). Statistically significant deviations from the null model between 15 and 10 ka are indicated by the red (busts) and blue (booms) rugs at the base of each panel. Time series of δ[18]O values from the NGRIP ice core (**d**), $CO_2$ from the Dome C, Antarctica, ice core (**e**), and insolation seasonality (**f**) are provided for comparison. Data sources are in Methods

**Trends in the Great Lakes**. In the Great Lakes region, mastodon, mammoth, and human SPDs suggest growth consistent with the null models through both the B-A and Clovis periods until population busts for mastodon and humans occur synchronously at 12.6 ka during the early YD (Fig. 4)—the mammoth bust to extinction occurs later at 11.9 ka. Highly significant positive relationships exist in this regional sample between the human SPD and those from both mastodon ($r_s = 0.54$, $P < 0.0001$) and mammoth ($r_s = 0.29$, $P < 0.0001$).

## Discussion

Insofar as these trends in the radiocarbon records accurately reflect the timing of significant population busts and the relationships between human and megafaunal populations, they allow us to consider taxon-specific probable causes for extinctions in the different geographic areas. These analyses suggest that a taxonomically and regionally diverse set of causal factors drove terminal Pleistocene megafauna extinctions. Since all taxa exhibit population growth consistent with the null models through the pre-Clovis B-A with busts leading to extinction occurring either during Clovis times or the YD, we focus on the period covered by the latter phenomena in addressing the causes for extinctions. The cause for mammoth, horse, and saber-toothed cat extinction on the contiguous US scale is most consistent with the activities of Clovis hunters (Fig. 5; Table 1). This is so because the demographic busts occur in Clovis times and strong negative relationships exist between the human−mammoth, human−horse, and human−saber-toothed cat SPDs. The human impacts on these taxa may have resulted from a variety of factors,

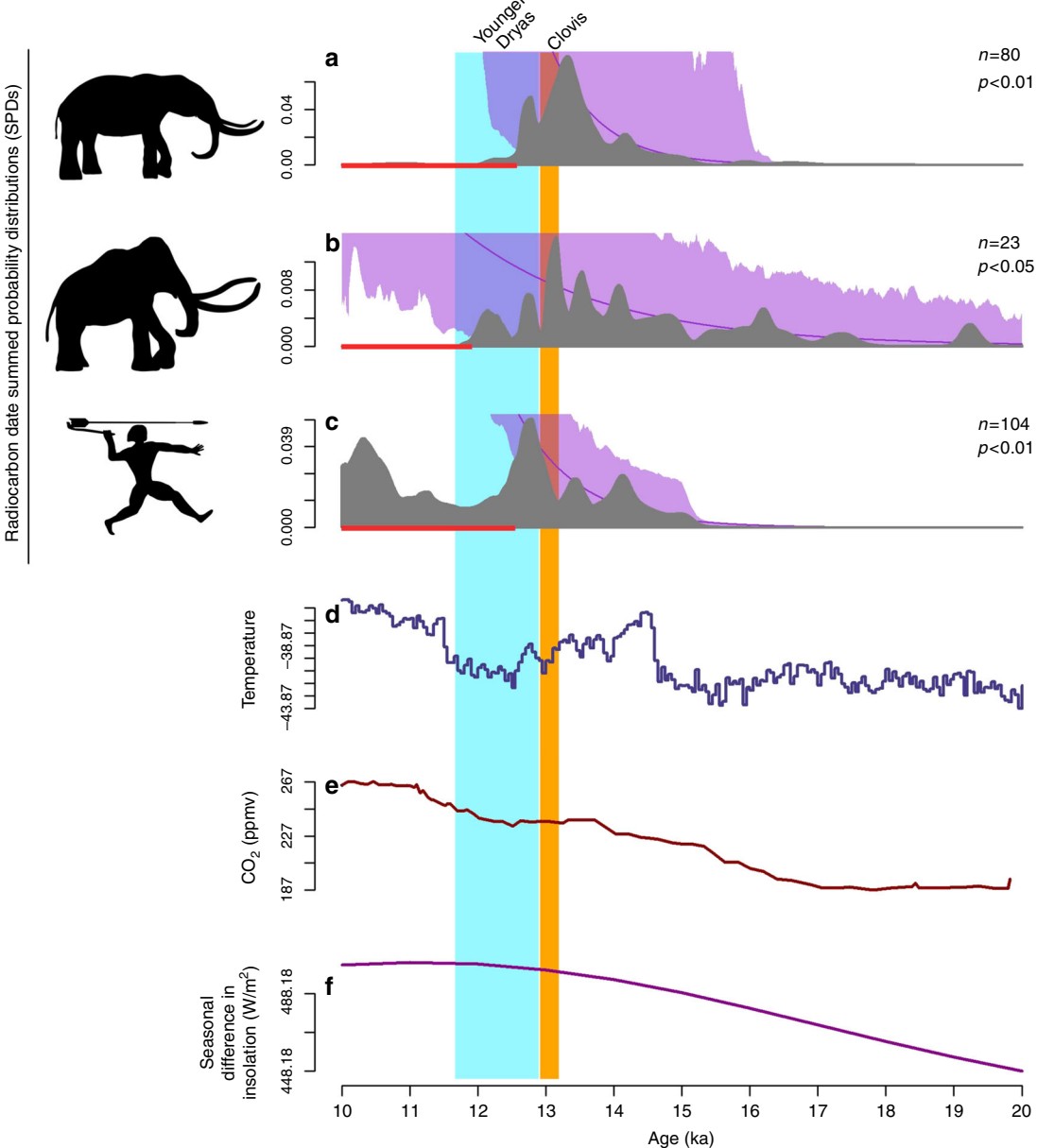

**Fig. 4** Megafaunal and human SPDs through time for the Great Lakes. Mastodon (**a**), mammoth (**b**), and human SPDs (**c**) are indicated in gray. The null model of exponential growth is indicated by the purple line; the purple shading denotes the 95% confidence interval around the null model (see Methods). Statistically significant deviations from the null model between 15 and 10 ka are indicated by the red (busts) and blue (booms) rugs at the base of each panel. Time series of $\delta^{18}O$ values from the NGRIP ice core (**d**), $CO_2$ from the Dome C, Antarctica, ice core (**e**), and insolation seasonality (**f**) are provided for comparison. Data sources are in Methods

such as direct hunting, as in the case of mammoth and horse, or indirectly as in the case of saber-toothed cat whose prey may have been depressed by human foragers. However, in no case does Shasta ground sloth or mastodon extinction appear linked to human hunting during the Clovis period or after given the lack of relationships between the sloth–human and mastodon–human SPDs and with their final busts to extinction occurring during the YD. The Great Lakes region provides no support for a causative role of Clovis or post-Clovis hunting in the extinction of either mastodon or mammoths—instead the extinctions in this region are consistent with YD climate change. The Southwest region provides the single case that is consistent with a multi-causal hypothesis—the mammoth and human SPDs in this region are negatively correlated but the terminal decline to extinction occurs

well-after Clovis toward the end of the YD. Post-Clovis foragers are thus implicated as playing the human role in mammoth extinction in the Southwest. Thus, three cases appear most consistent with Clovis hunting impacts, five imply YD climate change, and one is suggestive of a multi-causal scenario (Table 1).

This diversity of extinction causes is also generally consistent with the taxonomic composition of the small sample of sites (n = 15) that document a clear human role in the utilization of megafaunal taxa. Most notably, 80% of those sites contain the remains of mammoth and no site has yet to produce unequivocal evidence for the human use of Shasta ground sloth or any other ground sloth[32]. This is all the more remarkable given that Shasta ground sloth was among the most abundant megafaunal taxa on the terminal Pleistocene landscape, third only to mammoth

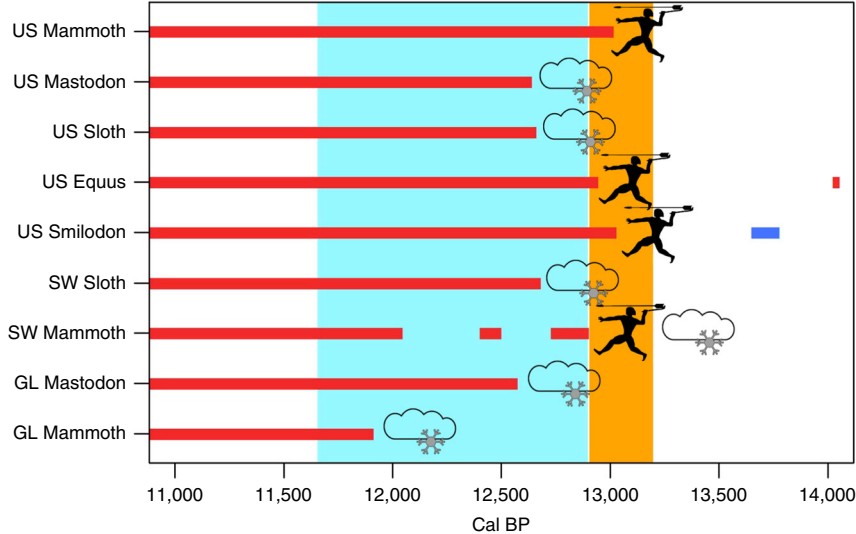

**Fig. 5** Summary of the chronology for megafaunal population busts (red lines) by region in relation to the Clovis period (orange column) and the Younger Dryas (light blue column). Symbols to the right indicate extinction cause(s) suggested by this analysis: clouds and snowflake = climate; human with spear = hunting

**Table 1 Summary of the timing of terminal population busts and suggested causes for megafaunal extinctions by taxon and region**

| Taxon by region | Dates (n) 10–20 ka | Date density (dates/yr) 11–15 ka | Date of terminal bust (cal BP) | Rank-order correlation with human SPD 11.7–15 ka (in all cases, $P < 0.01$) | Extinction consistent with: Clovis hunting[a] | Extinction consistent with: Climate change[b] | Extinction consistent with: Multicausal[c] |
|---|---|---|---|---|---|---|---|
| Contiguous US | | | | | | | |
| Mammoth | 74 | 0.013 | 13,014 | −0.590 | Yes | No | No |
| Mastodon | 99 | 0.023 | 12,639 | −0.045 | No | Yes | No |
| S. ground sloth | 40 | 0.010 | 12,659 | 0.353 | No | Yes | No |
| Horse | 25 | 0.005 | 12,944 | −0.434 | Yes | No | No |
| Saber-toothed cat | 21 | 0.002 | 13,028 | −0.663 | Yes | No | No |
| Human | 938 | 0.158 | — | — | — | — | — |
| Southwest | | | | | | | |
| S. ground sloth | 29 | 0.007 | 12,680 | 0.663 | No | Yes | No |
| Mammoth | 27 | 0.005 | 12,044 | −0.782 | No | No | Yes |
| Human | 153 | 0.025 | — | — | — | — | — |
| Great Lakes | | | | | | | |
| Mastodon | 80 | 0.019 | 12,573 | 0.543 | No | Yes | No |
| Mammoth | 23 | 0.004 | 11,912 | 0.293 | No | Yes | No |
| Human | 104 | 0.018 | — | — | — | — | — |

[a]Consistent with Clovis hunting (directly or indirectly) if declines to extinction begin during Clovis times and significant negative relationships occur between humans and megafauna
[b]Consistent with YD climate change if declines to extinction begin in YD absent negative relationships between humans and megafauna
[c]Consistent with multicausal hypotheses if significant negative relationships occur between humans and megafauna and declines to extinction begin during the YD

and mastodon to judge from the directly dated radiocarbon record. Moreover, only one of these kill/scavenging sites, the Pleasant Lake (MI) mastodon, occurs in the Great Lakes region where again our analysis provides no suggestion for a human role in megafaunal extinctions.

The geographic and taxonomic diversity in the causes for megafaunal extinctions implied by these data also underscores the utility of approaching the extinctions issue, where possible, through a focus on individual taxa in specific geographic settings. Although we were constrained in our ability to do this here by the currently available sample of directly dated megafauna—and this represents an important trade-off in conducting spatially focused analyses in general—this approach has long been adopted in the context of exploring megafaunal extinctions in northern Europe and Eurasia. Those analyses have shown that extinctions were staggered through space and time with some aligning closely with

climate and vegetational change but others appearing more consistent with human hunting[1–3,21]. Although still limited, recent work in North America has also focused on the assembly of high-resolution records of the biogeographic histories of individual taxa across smaller spatial scales through the integration of paleoclimatic, paleoecological, and archeological data[28–30,47,48,59].

Recent analyses of the biogeographic histories of mammoth and mastodon in the upper Midwest and northeast regions of North America[29,47,48] are most notable in this context as they each encompass parts of the broader Great Lakes area examined here. These studies adopt different methodologies for estimating population declines and extinctions and draw upon different geographic areas and samples of dated materials thus precluding detailed comparisons with our results. Still, the extinction windows suggested for mastodon in the upper Midwest and the New York

regions are 12.8–12.5 ka, and 12.5–11.9 ka, respectively[47,48]. Previous analysis of the collective megafaunal sample from a study focused on the New England states provided a similar range of 12.7–12.5 ka[29]. Our estimate for the final mastodon population busts for both the Great Lakes and for the contiguous US falls at 12.6 ka, within, or very close to, the date range of these previous estimates that are based on materials derived from smaller spatial scales and, as a result, smaller sample sizes.

Yet, despite similar extinction chronologies and broadly similar paleoclimatic, paleovegetational, and archeological histories, distinctive sets of causes are proposed to underlie extinctions in these different but overlapping regions. These range from dominant roles afforded to human hunting[48] or paleoclimatic change[29], to considerations of broader ecological processes and prior changes in the mammalian communities that occupied these regions[47]. Widga et al.[47], for example, argue that the upper Midwest proboscidean extinctions were related to a shift from top-down to bottom-up mechanisms of population regulation that resulted from earlier pre-Last Glacial Maximum extinctions and population declines of large carnivores. Uncontrolled by predation, terminal Pleistocene proboscideans in this region approached carrying capacity and were thus unstable and susceptible to climate-based population cycling and extinction. Although robust tests of this specific scenario have yet to be conducted, the results from our Great Lakes analyses are not inconsistent with it insofar as we detect no human role in the extinction process in this area, the significant demographic busts that lead to extinction for mastodon and mammoth occur during the YD, and none of our collection of directly dated large carnivores ($n = 134$) are from specimens derived from the Great Lakes region (Supplementary Data 1).

These analyses emphasize that a focus on individual taxa does not imply that the processes driving their extinction were unrelated to the mammalian communities of which they were a part. And although we have emphasized the climatic-environmental conditions that may have contributed to the significant megafaunal declines during the YD, other factors intrinsic to the large mammal community itself may have also played a role in the differential extinction that occurred during this time as many megafauna taxa, of course, survived this wave of extinctions. Most notably, ruminant taxa including bison (*Bison*), sheep (*Ovis*), deer (*Odocoileus*), elk (*Cervus*), and moose (*Alces*), that can more efficiently detoxify and utilize plant nutrient pulses in shorter growing seasons, survived disproportionately at the expense of monogastric taxa (e.g., mammoths, sloths, horses) that possess a unique digestive anatomy and have more conservative somatic growth rates[39]. And while many of the surviving megafauna taxa experienced body size diminution consistent with nutritional stress[29,39,50], and may have persisted at low densities during the terminal Pleistocene and early Holocene[41], a radiocarbon-based population history for *Bison* suggests that their populations grew consistent with an exponential null model from 15.0 ka through the Clovis period and until the end of the YD. In addition, and unlike any other megafauna taxa examined here, *Bison* also exhibit a population boom during the early YD (Fig. 6). Relatively recent immigrants from Eurasia, *Bison* are aggressive, highly fecund, herd-dwelling ruminants that experienced a substantial wave of dispersal in North America since the last interglacial and may have competitively excluded other megafaunal taxa, especially taxa already compromised by nutritional stress or impacted by human hunting[60–63]. Yet despite this successful late Pleistocene expansion, *Bison* also experienced significant body size diminution in several pulses, the first occurring sharply during the YD[62]. This trend may also be linked to declines in foraging efficiency, since it occurs in the absence of a change in age structure—which suggests the trend is unrelated to hunting

pressure—and during a period of reduced overall temperature that might otherwise select for increasing body size[62].

Although this study is based on the largest assembled collection of directly dated North American megafauna, we hope that it will stimulate future work to expand this radiocarbon record to allow the generation of higher-resolution demographic histories and more robust evaluations of these emerging patterns. It is worth noting again that this analysis applies to the subset of extinct Pleistocene mammalian genera for which we currently have a substantial record of direct radiocarbon dates and we still lack such dates for 18 of them. Insofar as relative date frequencies track population sizes, these taxa that lack a radiocarbon record may have been characterized by very small populations or were already extinct by the terminal Pleistocene. As in the case of mastodon extinction in eastern Beringia that clearly dates to near or beyond the ~40 ka limit of $^{14}$C dating[59], the cause for these apparent earlier losses must of course be found in factors other than YD environmental change or human impacts. Moreover, the degree to which the list of terminal Pleistocene megafaunal extinctions is shortened, the magnitude of this extinction phenomenon is brought closer in line to those that characterized previous interglacials.

## Methods

**Compiling radiocarbon datasets**. We compiled 521 direct radiocarbon dates on extinct megafauna derived from published sources for the contiguous US including a small sample ($n = 17$) of specimens derived from the southern margins of directly adjacent Canadian provinces (i.e., Alberta, Ontario, and Nova Scotia; Fig. 1; Supplementary Data 1). We included only direct AMS or conventional radiocarbon assays on hide, hoof, dung, and bone collagen; we excluded dates on apatite, whole bone, or bone mineral. In this comprehensive megafauna radiocarbon dataset we included genera that went extinct in North America as well as several well-defined extinct species that belong to genera that live on in the region, namely American lion, dire wolf, and Harrington's mountain goat. So as not to oversample megafauna during the period they overlap with people, we followed Surovell et al.[14] and excluded dates derived from unequivocal archeological contexts. To this end, we followed the vetting of Grayson and Meltzer[32] and excluded megafaunal dates from sites that represent clear kill and/or scavenging associations.

To obtain archeological radiocarbon dates, we queried the CARD[58] for all dates falling within the study region (south of 52 °N latitude) between 15.0 and 10.0 ka, the former chosen as the most widely accepted earliest date for the initial colonization of the Americas (following Weitzel and Codding[64]). Given that we made use of minimally vetted and user-supplied radiocarbon data from the CARD, it was necessary to ensure data quality prior to our analysis. We followed the approach of Weitzel and Codding[64] in removing all dates which are (1) duplicated in the database (based on the laboratory number), (2) marked as "anomalous" by the uploader, and (3) reported to originate from paleobiological or geological (i.e., nonanthropogenic) contexts. Additionally, we removed radiocarbon dates ($n = 54$) derived from megafauna skeletal material that originated from sites where convincing kill/scavenging associations could not be made[32] and thus would not relate to past human activity. This left us with a sample of 938 anthropogenic radiocarbon dates.

Although previous analyses using the CARD dataset for the Paleoindian period have taken the data at face value while recommending caution[23], the contested nature of pre-Clovis (>~13.2 ka) occupations warrants further scrutiny. We thus conducted separate analyses using the CARD dates between 13.2 and 10.0 ka but for the pre-13.2 ka period, only dates from the few most widely accepted pre-Clovis sites that have produced substantial radiocarbon records; these include Paisley Caves, Meadowcroft Rockshelter and Page-Ladson[65–67] (see Supplementary Note 1: Analysis of Vetted Pre-Clovis Dataset, Supplementary Tables 1 and 2, and Supplementary Figure 1). The results and conclusions based on these analyses do not differ substantively from those presented above using only the dates obtained from the CARD. As it has been argued that variation in economic complexity can influence per capita energy consumption and the production of dateable archeological materials (bone, charred wood, etc.), semi-independent of population size[68], we observe that our archeological date samples were derived from human populations that shared broadly similar technologies, foraging adaptations, and levels of economic complexity.

As noted above, our radiocarbon datasets for extinct megafauna are based exclusively on directly dated specimens derived from paleontological contexts following the systematic evaluation of Grayson and Meltzer[32]. Since no such evaluation has been provided for the *Bison* radiocarbon record, our analysis of this taxon (Fig. 6) is necessarily based on the collective assemblage of dates derived from both archeological and paleontological contexts. Since the radiocarbon dates for *Bison* are, to a large degree, a subset of the broader human radiocarbon record,

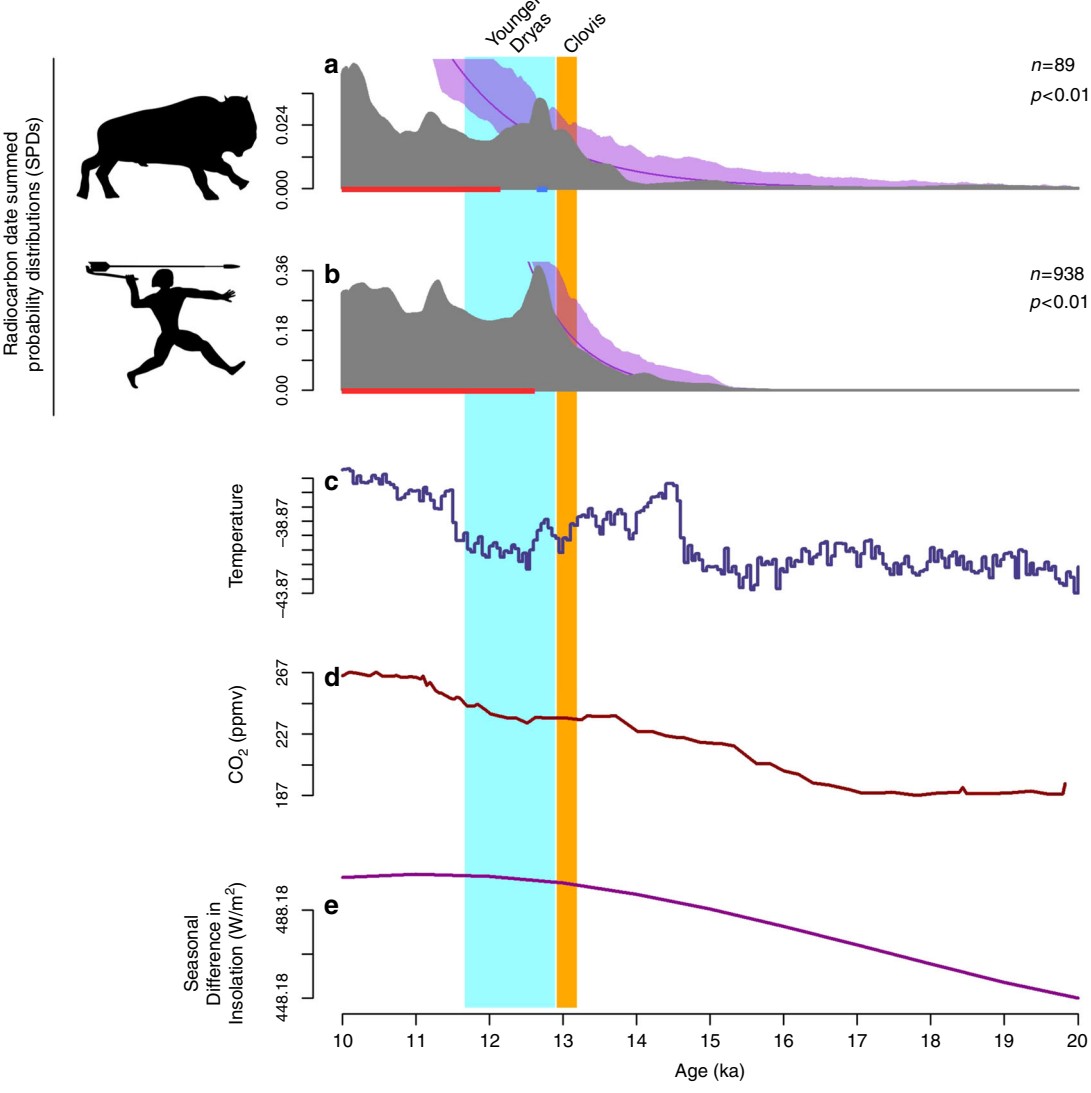

**Fig. 6** *Bison* and human SPDs through time for the contiguous United States. SPDs for *Bison* (**a**), and human populations (**b**) are indicated in gray. The null model of exponential growth is indicated by the purple line; the purple shading denotes the 95% confidence interval around the null model (see Methods). Statistically significant deviations from the null model between 15 and 10 ka are indicated by the red (busts) and blue (booms) rugs at the base of each panel. Time series of $\delta^{18}O$ values from the NGRIP ice core (**c**), $CO_2$ from the Dome C, Antarctica, ice core (**d**), and insolation seasonality (**e**) are provided for comparison. Data sources are in Methods

a bison−human correlation analysis would be inappropriate and, thus, was not conducted. The sample of direct dates for *Bison* ($n = 89$) was compiled from the CARD[58], Shapiro et al.[19], and McDonald[60].

**Constructing summed probability distributions**. Using the radiocarbon date samples obtained for both megafauna and humans, we first calibrated all dates based on the IntCal13 calibration curve using the *rcarbon* package[69] available for R. Calibrating dates using this package produces identical results to conventionally used software such as OxCal[54–57]. Using the *rcarbon* package, we then constructed SPDs for the contiguous US using a 200-year moving average from 10 to 20 ka for humans ($n = 938$) and all taxa represented by more than 20 dates for this period. These taxa include mammoth ($n = 74$), mastodon ($n = 99$), Shasta ground sloth ($n = 40$), horse ($n = 25$), and saber-toothed cat ($n = 21$). To examine megafaunal −human population relationships on more regional scales we derived separate SPDs for the Southwest and the Great Lakes regions as they are represented by the largest samples of megafaunal dates. We used a broadly encompassing definition of the Southwest here to include the states of California, Nevada, Utah, Arizona, and New Mexico; the Great Lakes region, for our purposes, includes all states and provinces that border any of the Great Lakes (i.e., Ontario, New York, Pennsylvania, Ohio, Michigan, Indiana, Illinois, Wisconsin, and Minnesota). For these two regions, we separately examined SPDs for the megafauna taxa represented by more than 20 dates, namely, Shasta ground sloth ($n = 29$) and mammoth ($n = 27$) in the Southwest, and mastodon ($n = 80$) and mammoth ($n = 23$) in the Great Lakes. For the period of greatest interest here (15.0−11.0 ka), date density (dates/year) ranges

from 0.158 to 0.002 for the different regional megafauna and human date records with a mean density value of 0.023. These values fall within the range of most published analyses using SPDs to reconstruct population trends[23–28,54–57]. Furthermore, Timpson et al.[55] determined that the method used here to construct SPDs is robust to small sample sizes. Randomly subsetting a regional radiocarbon SPD from Europe, they found that the shape of the curve was generally maintained with sample sizes as low as 12 dates spanning 4000 years (13% of the 93 dates used to construct their original SPD). However, while the general patterns of significant deviation from the null model were maintained in this small subset of dates, the exact timing of these deviations is susceptible to minor changes due to the widened confidence intervals associated with smaller sample sizes. SPDs constructed from small samples of dates may still therefore record accurate patterns.

**Significant SPD deviations from the null models**. We followed the approach of Shennan et al.[54–57] and compared each SPD to a null model to identify periods of time where the SPD deviates significantly from expected values. The null model to which we compared each SPD represents the trajectory each megafaunal taxon or human SPD followed prior to its decline. To create such a null model, we first used segmented regression analysis to fit paired quasi-Poisson family generalized linear models (GLMs) with a log link function to the SPD using the *segmented* package[70]. This analysis identified "breakpoints" at which the two GLMs were joined. These breakpoints represent the date at which the SPD shifts from an overall pattern of increasing to decreasing values. We next created a single quasi-Poisson family GLM with a log link that was based on the SPD values from 20.0 ka to the taxon's

specific breakpoint date, thereby representing only the increasing portion of the SPD. To generate the values of the null model itself, we allowed this GLM to predict values beyond the taxon's breakpoint date based on the pre-breakpoint SPD trajectory. The resulting null model therefore reflects the pre-breakpoint (and pre-decline) exponential growth rates of each SPD, or the expected trend of the SPD should it never decline.

Then, using the modelTest function in the *rcarbon* package, we used a Monte-Carlo method to simulate 500 SPDs based on this null model and generate a confidence interval to which we compare the original SPD. This was done by taking the number of dates used to construct the original SPD and distributing them proportionally under the null model, using the shape of the null model to provide probability estimates for each year according to which dates were simulated. These simulated dates were then "uncalibrated" by passing them back through the IntCal13 curve: for each simulated calendar date, a random value was selected from within the calibration curve's error range for that given year. Standard deviations associated with uncalibrated dates were next sampled from our original dataset of $^{14}$C dates with replacement and attached to each simulated date, ensuring that the simulated dates have comparable error estimates to the originals. The simulated $^{14}$C dates were then recalibrated using these sampled error values and used to construct an SPD. This SPD was created using a 200-year moving average applied to each year between 10.0 and 20.0 ka. Such a moving average helps to control for any small-scale artifacts of the calibration curve that would otherwise show through in the resulting SPD. The SPD was next locally transformed using Z-scores to standardize the probability distribution and make it comparable to others, which also has the effect of "de-trending" the resulting SPD so that any features of the original SPD and the calibration curve were removed. We repeated this process 500 times and a 95% confidence interval was generated around the exponential null model based on the Z-scores of each SPD, making it possible to identify time periods for which the observed SPD significantly deviates from the null model. Finally, a "global" P value expressing the overall deviation of the entire observed SPD from the null model was generated by calculating the proportion of simulated SPDs for which the total area lying outside the 95% confidence interval was as or more extreme than that of the observed SPD. These steps were then repeated for each taxon evaluated in each of the three regions, as well as for all taxa from all regions using taphonomically corrected dates following Surovell et al.[71] (see Supplementary Note 2: Analysis of Taphonomically Corrected Data, Supplementary Table 3, and Supplementary Figure 2–5).

**The utility of comparisons to an exponential null model.** Shennan et al.[54,55] emphasize that this approach, based on an exponential null model, provides two key benefits. First, using an exponential null model provides a hypothesis of unregulated growth against which to test an SPD: a null hypothesis in which the population growth rate never declines. Of course, real-world populations are rarely unregulated, making a logistic growth model a generally more realistic representation of population dynamics. However, applying a logistic growth model to these data present challenges. Logistic population growth models reach their top asymptote due to the effect of carrying capacity, which constrains a population from growing beyond a certain point typically dictated by food supply. It is not possible to derive the terminal Pleistocene carrying capacity for each of these megafauna taxa, making it problematic to assign such values in constructions of null models. We do not know the available food supply for each taxon, nor should we assume that each taxon had reached its carrying capacity prior to its declines towards extinction. Nonetheless, to ensure that our choice of exponential null models does not alter our results in any substantial way, we repeated all analyses using logistic null models (see Supplementary Note 3: Analysis Using Logistic Null Models and Supplementary Figures 6–8). These models were fit using maximum likelihood estimation of five-parameter logistic models based on Richard's Equation. A maximum likelihood approach permits the estimation of the carrying capacity parameter (and all other parameters) based on the characteristics of the observed data, not any a priori expectation. Such values may not reflect real-world carrying capacity, for example, but are an empirically robust way of fitting such a model to these data in the absence of other information. Attempts to fit the more commonly known three-parameter logistic model typically failed as most of these SPDs behave exponentially prior to their breakpoint dates, not logistically. Many of the logistic models fit here are therefore either poor fits to these data or obtain marginally better fits at the cost of several additional degrees of freedom. For this reason, these null models frequently resemble their exponential counterparts because maximum likelihood estimation places the top asymptote (i.e. carrying capacity) far above the SPD in order to optimize the model fit to the available data.

Second, an exponential model increasing towards the present takes the same form as the taphonomic correction curve first proposed by Surovell et al.[71] and revised by Williams[25]. Taphonomic loss of dateable organic materials increases with time according to an exponential function; therefore, deviations of an SPD from an exponential null model are, in effect, deviations from a taphonomic correction curve. Shennan et al.[54–57] therefore suggest that dates for which SPD values are significantly greater than the exponential null model represent not only higher populations than would be expected given unregulated growth (i.e., "booms"), but also greater representation of dated carbon than would be expected according to a taphonomic correction curve. Conversely, when SPD values are significantly lower than the null model, this reflects lower populations than would

be expected (i.e., "busts") as well as a dearth of dated carbon relative to the amount expected given the taphonomic correction curve. However, because this approach simply provides an analog for the taphonomic correction curves of Surovell et al.[71] and Williams[25] against which to compare an SPD, and does not actually correct the SPD itself, we also repeat all of our analyses using taphonomically corrected dates, following the equation provided by Surovell et al.[71]. No substantive differences are apparent between the analyses we presented above and those based on the taphonomically corrected data (see Supplementary Note 2).

**Potential sources of error.** Bias in researcher submission of samples to date is also a potential source of error in both the CARD and megafaunal datasets and may be especially concerning in the context of the Paleoindian period in general (including Clovis, Folsom and related fluted point complexes) that is associated with heightened interest among archeologists. Yet we also anticipate equal or greater interest and bias in date submission for potential pre-Clovis sites. Collectively, we thus expect some oversampling for the Pleistocene-Holocene transition (~15.0–10.0 BP) period in general but see no mechanism to account for submission bias and oversampling to fluctuate in a patterned way within it. It is also unclear how researcher submission bias could account for the trends in the paleontological megafauna database; there would seem to be no less inherent interest in a mammoth skeleton (or a piece of Shasta ground sloth dung) from geological/stratigraphic contexts suggestive of the YD period, compared to those from contexts immediately preceding it. And as noted above, we excluded from our megafaunal dataset those dates derived from clear archeological contexts to avoid the inflation of megafauna numbers during the period they overlapped in time with people, while also only including one date per megafauna individual.

**Paleoclimatic data.** The paleoclimatic data presented in Figs. 2–4, 6 (and Supplementary Notes 1–3) are derived from the $\delta^{18}$O values from the NGRIP ice core[72], the $CO_2$ data from the EPICA Dome C, Antarctica, ice core[46], and the insolation seasonality values at 60°N[45].

**Code availability.** The R code used in this study is provided in Supplementary Data 2.

## Data availability

All megafauna radiocarbon data used are from previously published sources and are listed in Supplementary Data 1. All human radiocarbon data are provided from the CARD: Canadian Archaeological Radiocarbon Database (CARD 2.1) Geospatial Radiocarbon Data (Accessed 15 May 2017). See http://www.canadianarchaeology.ca/. A reporting summary for this Article is available as a Supplementary Information file.

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

## Acknowledgements

We thank Brian Codding, Duncan Metcalfe, Kristen Hawkes, Isaac Hart, Nicole Herzog, Allison Wolfe, and the University of Utah Archaeological Center lab group for helpful comments on the manuscript and Isaac Hart for help with the figures.

## Author contributions

Both authors contributed equally to the design of the study, analysis and interpretation of the data, and the writing of the manuscript.

## Additional information

**Competing interests:** The authors declare no competing interests.

