## [Peer Review File · Nature Communications]

Reviewers' Comments:

Reviewer #1:

Remarks to the Author:

Overall, this is an excellent and innovative analysis of interest to a wide variety of disciplines (archaeology, paleontology, ecology). This manuscript examines the important issue of human and megafauna population dynamics during the late Pleistocene. Its overall finding is that cause of the extinction of Pleistocene fauna occurred for various reason, including human impacts and climate change, although depending on the species or the region of North American the relative importance of these causes differed. The analysis applied an accepted methodology of using summed calibrated radiocarbon date distributions and simulations to reconstruct population levels of megafauna and humans. While over the last several years there has been criticisms of type of statistical methodology used in this study, the field has become growingly comfortable with the type of analyses performed in this manuscript. By and large the current authors follow established and appropriate methods and the results are reasonable and acceptable. It deserves to be published after some modest revisions to the structure of the analysis and presentation. Below I summarize my substantive criticism as well as minor text-based edit suggestions.

My primary criticism concerns the authors' poor decision to group the majority of megafauna in the analysis into an "all other" category. As discussed in pages 8 and 9, there are interesting (and perhaps contradictory) results found with the "all other megafaunal" dataset. One group with a large number of species obscures a huge amount of potentially important detail. Why not produce separate SPDs for commonly represented taxa, such as *Arctodus*, *Camelops*, *Canis dirus*, *Equus*, *Oreamnos harringtoni*, and *Smilodon*? As stated in the text, the authors recognize the methods used here can construct reliable SPDs with samples as small as 12 dates. All of these taxon are represented by at least 16 dates. These species therefore should be analytically viable units of analysis. More importantly, the extinctions (whether the result of human impacts, climate change, or some combination of effects) occur at a species level (see Grayson 2007, 2016) and grouping taxa together is contradictory to this perspective. Combining quite distinctive species also obscures our potential to learn about how population dynamics differ among species with different adaptive strategies or with potentially different relationships with humans. For example, humans hunted *Equus*, *Camelops*, *Cuvieronius* (which is missing from this analysis), *Mammot*, and *Mammuthus* (e.g., Grayson and Meltzer 2015). Humans may also have hunted *Camelops* and *Oreamnos harringtoni* (although I do not believe there is good evidence to support this idea at present). In contrast, direct human predation is probably not a reasonable explanation for the loss of *Arctodus*, *Canis dirus*, and *Smilodon*. Because these species have potentially different "trajectories" towards extinction, in my opinion, it is critical to understand their unique population dynamics.

A species-level focus is so critical that I would recommend that the authors drop from consideration any taxon for which they do not have adequate samples, and instead acknowledging that at present we probably only have adequate data to consider particular species. That means the loss of a large number of species from consideration (e.g., *Bootherium*, *Castoroides*, *Cervalces*, etc). I say this even though I feel that the information presented in Supplementary Table 1 is a huge contribution to the field. While I feel it is critical to consider species like *Arctodus*, *Camelops*, *Canis dirus*, *Equus*, *Oreamnos harringtoni*, and *Smilodon* individually, if the authors felt it necessary to keep the remaining lesser represented species in the analysis it could be done by using a small number of "other" categories that are based are reasonable biologically-informed groups, such as herbivores vs carnivores, grassland or forest species, or some other groupings.

On a related note, the paper suffers from only considering species that went extinct. While reading this manuscript, I continually wondered how these trends compare to trajectories of species that

survived. On pages 10 and 11, the authors note that certain megafauna taxa (Bison, Ovis, *Odocoileus*, and Alces) that survived the Pleistocene extinction had different digestive adaptive capacities or were highly plastic in body size, but I think it would be important to also examine the nature of their population dynamics compared to those species that went extinct. These four species were unquestionably heavily preyed upon by Clovis and later hunters. How did their populations change at the time of the appearance of humans, and did they have a different trajectory? It would be interesting to identify population trajectory for the surviving taxon and compare them to the Null models use for the extinct species. If it would be difficult to collect the data for all surviving species, perhaps just include one species, such as bison, and collect a large sample of dates for this taxa between the 10-20 ka time frame considered in this analysis.

I also have a series of comments or suggestions related to minor, text-specific statements, which are discussed below:

- Page 3, paragraph 1: there are a series (sentences 2 and 4; also on page 5) of direct or indirect references to publications related to a Pleistocene extraterrestrial impact. This hypothesis has been widely discredited (e.g., Holliday et al. 2016; Meltzer et al. 2014), and I do not believe that it is necessary at this time to even acknowledge this hypothesis. If it is mentioned (as with hyperdisease), clearly state that there is very limited support for this hypothesis.
- Page 3, paragraph 1: You state that this paper is somewhat unique in its focus on the megafaunal population dynamics in the context of human predation and changing climates. While I agree with this statement in a general sense, it seems to me that other researchers (Ugan and Byers 2007, Boulanger and Lyman 2014) have considered this issue in perhaps slightly different ways. It would be appropriate to acknowledge prior work on this topic or at least emphasize how this articles makes significant methodological improvements in addressing the topic.
- Page 4, paragraph 2: the sentence that discusses the contrasting explanation for why there are a paucity of kill sites associated with extinct faunal needs some revision. First, the references used in this sentences are generally quite either old (Martin 1981, Mosimann and Martin 1975) or secondary sources citing older works which does not support overkill at all (Grayson and Meltzer 2015) . As such I do not think this topic has as much relevance as it once did. Recent supporters of overkill do not seem to rely on the argument that the extinction event occurred too fast to be preserved archaeologically (although see Surovell and Grund 2012 as a more recent citation on this basic issue. In addition, recent critics of overkill (e.g., Meltzer 2015, Hill et al. 2008) have started to emphasize that surviving taxa, which have been unquestionably preyed upon by humans, are very visible archaeologically event during the period of extinction which makes the paucity of extinct species seems more unusual.
- Page 12, paragraph 3: the statement about dates from “few widely accepted pre-Clovis sites” has to be supported with citations. There are proponents and critics of almost all of these sites, and so it would be important to reference citations that demonstrate that these particular sites are in fact widely accepted.
- Page 13, paragraph 2: there appears to be some contradiction on the number of available radiocarbon dates listed in the text and listed in Supplementary Information Table 1. For example, the Excel table provided indicates 96 samples of mammoths, 108 for mastodon, and 52 for Shasta ground sloths. Please check and confirm counts.
- Page 16, paragraph 2: In the section on potential sources of error, there is an overly vague statement about potential problems due to inaccuracy of radiocarbon dates “derived some time ago”.

However, this statement is so vaguely worded that it is not clear what actually is the problem with these dates and what time frame is implied from "some time ago". If the authors are referring to standard ^{14}C dates from the 1950s or 1960s there is probably a lot of agreement that these dates should not be used. . . but maybe not so much agreement about AMS dates from the late 1980s. It is not clear what is meant by this statement or to what extent this problem exists in this study. One way to clarify would be to provide information on the 521 radiocarbon dates collected from CARD as a supplementary table.

In summary, I believe with some modest revisions as described above, this manuscript can make a significant contribution to the field and should be of great interest to the readers of Nature Communications.

Reviewer #2:

Remarks to the Author:

This is an excellent paper, addressing a topic of wide interest with a clear set of hypotheses and an appropriate testing methodology. It is extremely clearly written and the methods used are very well explained and justified. The discussion is relevant and illuminating.

I was initially concerned that they did not appear to be addressing Surovell et al's proposed correction for using summed radiocarbon probabilities as demographic proxies but they do get it to in the end and show in the SI that it does not make any difference to their conclusions.

Reviewer #3:

Remarks to the Author:

This paper is greatly to be praised for its hypothesis-testing approach, its very careful treatment of data, its innovative methodology, and its balanced, thoughtful discussion. In this it stands in very favourable contrast to many recent papers on late Quaternary extinctions, which are based on questionable modelling of poor-quality data (and mostly seeking to promote an overkill model of extinction, whereas I find the present contribution admirably even-handed). I do, however, have some significant comments of a methodological nature that I hope the authors will be able to address. To make the review process easier for myself and the authors, I have added line numbers to the text in the attached version.

I have three major concerns and various more minor ones:

(1) The first major concern is about the a priori restriction of possible climate-induced causality to the Younger Dryas (YD). The whole paper makes this assumption and it is stated explicitly: "Climate-based causes would thus be suggested in cases where megafaunal population declines that culminate in extinction occur during the YD...". What is the justification for this? What about the LGM, the late-glacial interstadial (GI-1) or indeed the warming of the early Holocene, as well as the YD (GS-1)? Why might not these have impacted megafauna too? A priori there is no reason why the YD should be singled out as the only chronological correlation with megafaunal decline that will be accepted as causal. Especially since your analysis looks at the whole 20-10 ka period, it is simply not necessary to set this up as such a limited hypothesis in advance. Now, in the event, the authors' data do show the YD as correlated with decline in many species, and that's fine, so the YD as a climatic cause can and should be discussed *after* we have seen those results, but not as an a priori assumption.

(2) Analogously to the above, why are only the Clovis people tested as possible overkill perpetrators?

The date span goes to 10 ka, well beyond Clovis, but while Clovis is mentioned multiple times in the text there is nothing about the people in the post-Clovis interval. Even in Figure 2 only Clovis is indicated and any faunal changes after that time are attributed to climate. Yet the authors state that human populations over 5 millennia 'shared broadly similar technologies and foraging adaptations' (lines 315-318). In the first place, this statement needs further explanation and references. What cultures preceded and followed Clovis and how do you know they had similar adaptations etc? And if they had similar foraging adaptations, why might not they be implicated in the megafaunal declines & extinctions? This issue needs at the very least to be discussed.

(3) I found the conclusions less believable than they otherwise would have been because of the imposition of an exponential growth curve as the null hypothesis for megafaunal population size. Animal populations do not go on growing exponentially forever; the general model in population biology is a logistic, S-shaped curve in which initial exponential increase then levels off. If we look at the sloth in Fig. 2 for example, in the interval 15-13 ka it develops a very steep exponential increase – it seems wholly unrealistic to project that through the roof as the null model. Instead, a logistic curve would form a more appropriate and believable null for the authors' study. At the very least it should be added as sensitivity study. It is not as though most of these megafaunal species had just entered the continent as humans had; nor that they were expanding into the huge area vacated by the Laurentide ice sheet (because the bulk of your data is south of that). These considerations outweigh Shennan's supposed 'benefits' of an exponential model (line 379). Nor do I accept the Surovell model that preservation increases (and taphonomic loss decreases) exponentially, or in any regular manner at all, through the Late Pleistocene (lines 388-9). The Late Pleistocene is a very short interval relative to geological history, and climate-driven palaeoenvironmental effects on preservation potential (e.g. vegetation cover, frozen water), that we know to be strongly oscillating, are likely to be wholly dominant over a simplistic inverse chronological effect. I in any case felt muddled by the authors' account (lines 392-4) – is the modelled exponential increase in dates due to population growth, taphonomic effects, or some unknown combination of the two?

Minor comments by page/line number:

4/73-76: These conclusions of previous authors should be cited a little more critically. For example, radiocarbon data from S America is still very sparse, and even for N Am many extinct taxa have few or no dates.

5/106: 'these features converge' I found ambiguous. You mean 'several of these factors coincide'?

5/110: it is unclear what is meant by 'this interglacial cycle' in the context of a discussion about the Younger Dryas (YD). The YD is not a glacial but a stadial. Or do the authors mean the whole last glacial/interglacial cycle?

5/112-116: having acknowledged that there is geographic variation in the palaeoenvironmental effects of the YD, the authors should state specifically what is known of these changes in N Am S of the ice (their study area). And not just for the YD but for the whole GI1-GS1-MIS1 cycle.

6/152-3: for consistency, give all species full binomen.

7/155: does 'postdating 20.0 ka' mean 20-10 ka or 20-15 ka?

7/166: is the dating quantity and precision good enough to demonstrate 'a brief return to exponential growth' (or any other pattern) in a period of only 100 yr (12.8-12.7 ka)?

7/169-70: 'Significant negative correlations would be consistent with a human hunting impact...' – or just preferring different climates/habitats (the reciprocal of the inferred human-sloth relationship)?

7/173-175 (and more generally): For the whole continent, can you affirm that humans & megafauna were even in the same areas? If not, your tests for interaction would be invalid.

9/221: For extinctions in northern Eurasia it seems odd not to cite any of the numerous papers by Stuart & Lister who have done the most work in that region.

10/247: With no disrespect to Widga and colleagues, I was slightly bothered by the apparently

uncritical acceptance of their extinction scenario that is supported by hardly any dating of the carnivore extinctions or this improbable sounding mechanism of extinction through overpopulation. You say your data are 'consistent' with it but in fact they provide no evidence for this specific scenario – I would prefer if you added that!

10/256: 'formerly' with respect to what/when?

11/264: what is the relevance of bison being aggressive?

11/268-271. I found this passage unconvincing, resting on a string of doubtful assumptions. How is it known that foraging efficiency decreased? By exclusion? Is the change in age structure of the fossil assemblage real or taphonomic (or collecting bias)? And the idea that cold selects for large body size (Bergmann's Rule) is much debated.

11/278-9: As I am sure the authors realise, taxa may also lack a radiocarbon record because researchers tend to expend resources building chronologies of species that already have dates... or those species might have failed dating because all the bones are too old.

12/298: maybe 'arbitrary' is not quite the right word; I imagine it was carefully chosen as the most broadly acceptable earliest likely date for colonization. As an aside, the arguments over the date of 'initial colonization' can perhaps be placed to one side by focusing on the earliest significant colonization of the continent (and hence plausible impactors on megafauna).

12-13/315-318: As well as the issue about Clovis discussed above, I found this passage hard to understand. (1) "variation in energy consumption" – do you mean per capita energy consumption? (2) "a factor that can influence the relationship between carbon production and population levels" – I am afraid I couldn't understand what this meant. And (3) "can be held constant in this case" – please explain why is this significant?

13/334: recommend keeping dates in chronological order, so 15.0-11.0 ka.

16/403-4: The authors are to be praised for their careful approach to date auditing. But various other authors have discussed these issues so it would not be inappropriate to cite them – e.g. in relation to the point discussed here, Lister & Stuart (2013) applied a 1980 cut-off to exclude dates made before proper washing of collagen became routine.

There are also a very few minor suggested textual alterations in the attached version.

Dear Editor,

Please find our point-by-point response to the reviewer's comments below.

Jack M. Broughton and Elic M. Weitzel

Reviewer # 1.

Overall, this is an excellent and innovative analysis of interest to a wide variety of disciplines (archaeology, paleontology, ecology). This manuscript examines the important issue of human and megafauna population dynamics during the late Pleistocene. Its overall finding is that cause of the extinction of Pleistocene fauna occurred for various reason, including human impacts and climate change, although depending on the species or the region of North American the relative importance of these causes differed. The analysis applied an accepted methodology of using summed calibrated radiocarbon date distributions and simulations to reconstruct population levels of megafauna and humans. While over the last several years there has been criticisms of type of statistical methodology used in this study, the field has become growingly comfortable with the type of analyses preformed in this manuscript. By and large the current authors follow established and appropriate methods and the results are reasonable and acceptable. It deserves to be published after some modest revisions to the structure of the analysis and presentation. Below I summarize my substantive criticism as well as minor text-based edit suggestions.

My primary criticism concerns the authors' poor decision to group the majority of megafauna in the analysis into an "all other" category. As discussed in pages 8 and 9, there are interesting (and perhaps contradictory) results found with the "all other megafaunal" dataset. One group with a large number of species obscures a huge amount of potentially important detail. Why not produce separate SPDs for commonly represented taxa, such as *Arctodus*, *Camelops*, *Canis dirus*, *Equus*, *Oreamnos harringtoni*, and

Smilodon? As stated in the text, the authors recognize the methods used here can construct reliable SPDs with samples as small as 12 dates. All of these taxon are represented by at least 16 dates. These species therefore should be analytically viable units of analysis. More importantly, the extinctions (whether the result of human impacts, climate change, or some combination of effects) occur at a species level (see Grayson 2007, 2016) and grouping taxa together is contradictory to this perspective. Combining quite distinctive species also obscures our potential to learn about how population dynamics differ among species with different adaptive strategies or with potentially different relationships with humans. For example, humans hunted Equus, Camelops, Cuvieronius (which is missing from this analysis), Mammut, and Mammuthus (e.g., Grayson and Meltzer 2015). Humans may also have hunted Camelops and Oreamnos harringtoni (although I do not believe there is good evidence to support this idea at present). In contrast, direct human predation is probably not a reasonable explanation for the loss of Arctodus, Canis dirus, and Smilodon. Because these species have potentially different “trajectories” towards extinction, in my opinion, it is critical to understand their unique population dynamics.

A species-level focus is so critical that I would recommend that the authors drop from consideration any taxon for which they do not have adequate samples, and instead acknowledging that at present we probably only have adequate data to consider particular species. That means the loss of a large number of species from consideration (e.g., Botherium, Castoroides, Cervalces, etc). I say this even though I feel that the information presented in Supplementary Table 1 is a huge contribution to the field. While I feel it is critical to consider species like Arctodus, Camelops, Canis dirus, Equus, Oreamnos harringtoni, and Smilodon individually, if the authors felt it necessary to keep the remaining lesser represented species in the analysis it could be done by using a small number of “other” categories that are based are reasonable biologically-informed groups, such as herbivores vs carnivores, grassland or forest species, or some other groupings.

Response: We fully agree with this criticism and have modified the paper to a focused genus and species-level analysis. We now focus on the population reconstructions of the best represented taxa in our sample, specifically five taxa that are represented by more than 20 radiocarbon dates between 10-20 ka. These include mammoth, mastodon, Shasta ground sloth, horse, and saber-toothed cat. We have omitted the analysis of the “other megafauna” category.

On a related note, the paper suffers from only considering species that went extinct. While reading this

manuscript, I continually wondered how these trends compare to trajectories of species that survived. On pages 10 and 11, the authors note that certain megafauna taxa (Bison, Ovis, *Odocoileus*, and *Alces*) that survived the Pleistocene extinction had different digestive adaptive capacities or were highly plastic in body size, but I think it would be important to also examine the nature of their population dynamics compared to those species that went extinct. These four species were unquestionable heavily preyed upon by Clovis and later hunters. How did their populations change at the time of the appearance of humans, and did they have a different trajectory? It would be interesting to identify population trajectory for the surviving taxon and compare them to the Null models use for the extinct species. If it would be difficult to collect the data for all surviving species, perhaps just include one species, such as bison, and collect a large sample of dates for this taxa between the 10-20 ka time frame considered in this analysis.

Response: This is also an excellent suggestion and the revised version includes an analysis of an SPD-based late Pleistocene population reconstruction for *Bison*. See discussion on page 11 and Figure 6.

I also have a series of comments or suggestions related to minor, text-specific statements, which are discussed below:

- Page 3, paragraph 1: there are a series (sentences 2 and 4; also on page 5) of direct or indirect references to publications related to a Pleistocene extraterrestrial impact. This hypothesis has been widely discredited (e.g., Holliday et al. 2016; Meltzer et al. 2014), and I do not believe that it is necessary at this time to even acknowledge this hypothesis. If it is mentioned (as with hyperdisease), clearly state that there is very limited support for this hypothesis.

Response: We changed the text to read: *“Multi-causal arguments emphasizing the synergistic effects of both hunting and climate change are commonly advocated²⁻⁴ and other factors, such as hyperdisease¹¹ and extraterrestrial impacts⁵ have also been proposed but have little empirical support.”*

- Page 3, paragraph 1: You state that this paper is somewhat unique in its focus on the megafaunal population dynamics in the context of human predation and changing climates. While I agree with this statement in a general sense, it seems to me that other researchers (Ugan and Byers 2007, Boulanger and Lyman 2014) have considered this issue in perhaps slightly different ways. It would be appropriate

to acknowledge prior work on this topic or at least emphasize how this articles makes significant methodological improvements in addressing the topic.

Response: We changed the relevant text to underline the uniqueness of this study. The text now reads: *“Much of this work, and all of it focused on North America south of glacial ice, has approached the extinctions issue by reconstructing trends in the megafauna population as a single entity which may obscure significant species level detail and variability. We build on this work here in an analysis of late Pleistocene population histories for humans and five of the best-dated megafaunal species in North America south of glacial ice, including mammoth (Mammuthus sp.), American mastodon (Mammot americanum), horse (Equus sp.) Shasta ground sloth (Nothrotheriops shastensis), and saber-toothed cat (Smilodon fatalis). The study derives and tests mutually exclusive, quantitative implications for overkill, climate change, and multi-causal hypotheses for the extinctions of these taxa using a newly developed statistical approach that is robust to relatively small samples of dates”.*

- Page 4, paragraph 2: the sentence that discusses the contrasting explanation for why there are a paucity of kill sites associated with extinct faunal needs some revision. First, the references used in this sentences are generally quite either old (Martin 1981, Mosimann and Martin 1975) or secondary sources citing older works which does not support overkill at all (Grayson and Meltzer 2015) . As such I do not think this topic has as much relevance as it once did. Recent supporters of overkill do not seem to rely on the argument that the extinction event occurred too fast to be preserved archaeologically (although see Surovell and Grund 2012 as a more recent citation on this basic issue. In addition, recent critics of overkill (e.g., Meltzer 2015, Hill et al. 2008) have started to emphasize that surviving taxa, which have been unquestionably preyed upon by humans, are very visible archaeologically event during the period of extinction which makes the paucity of extinct species seems more unusual.

Response: We addressed this by adding the citation of Surovell and Grund (2012) and adjusting the text of this paragraph to read: *“Critics of overkill have long emphasized the small number of securely documented Clovis kill sites considering the magnitude of the slaughter implied^{1,12} but advocates counter that the available evidence is consistent with expectations given the narrow window of time over which the extinctions are believed to have occurred^{31,33}. Recent critics have also emphasized that surviving taxa, unquestionably harvested by human consumers, are highly visible archaeologically during the period of extinction, a fact that makes the paucity of extinct taxa in clear anthropogenic contexts all the more unusual³².”*

- Page 12, paragraph 3: the statement about dates from “few widely accepted pre-Clovis sites” has to be supported with citations. There are proponents and critics of almost all of these sites, and so it would be important to reference citations that demonstrate that these particular sites are in fact widely accepted.

Response: We added the following recent references from a range of high-profile authors with varying regional expertise that collectively suggest the three sites are among the most widely accepted pre-Clovis sites that have also produced substantial radiocarbon records.

66. Grayson, D.K. *The Great Basin: A Natural Prehistory* (University of California Press, Berkeley, 2011).

67. Anderson, D.G., Smallwood, A.M., Miller, S. Pleistocene human settlement in the southeastern United States: current evidence and future directions. *Paleoamerica* **1**, 7-51 (2015).

68. Waters, M.R., Stafford, T. The first Americans: a review of the evidence for the late Pleistocene peopling of the Americas. *Paleoamerican Odyssey* eds. Graf, K., Ketron, C., Waters, M. (Texas A&M University Press, 2014), pp. 543-562.

- Page 13, paragraph 2: there appears to be some contradiction on the number of available radiocarbon dates listed in the text and listed in Supplementary Information Table 1. For example, the Excel table provided indicates 96 samples of mammoths, 108 for mastodon, and 52 for Shasta ground sloths. Please check and confirm counts.

Response: The issue here is that we provide all available radiocarbon dates in the Supplementary Information Table 1 and we draw on this total sample of dates in our discussion of the relative date abundances for the different megafaunal taxa in the first paragraph of the results section (page 7). However, and as discussed in the text (page 13, paragraph 2) our analyses of the SPDs focus on selected taxa for the period *between 10 and 20 ka* so the numbers for the latter are different (lower) than the total sample of dates.

- Page 16, paragraph 2: In the section on potential sources of error, there is an overly vague statement about potential problems due to inaccuracy of radiocarbon dates “derived some time ago”. However, this statement is so vaguely worded that it is not clear what actually is the problem with these dates and what time frame is implied from “some time ago”. If the authors are referring to standard 14C dates from the 1950s or 1960s there is probably a lot of agreement that these dates should not be used. . . but maybe not so much agreement about AMS dates from the late 1980s. It is not clear what is meant by this statement or to what extent this problem exists in this study. One way to clarify would be to provide

information on the 521 radiocarbon dates collected from CARD as a supplementary table.

Response: To address this issue, we simply omitted the first four sentences in the *Potential Sources of Error* section (page 16, paragraph 2) that the reviewer found to be unclear. In retrospect, we are in fact very explicit elsewhere in the paper about the nature of the radiocarbon datasets we are using. For example, we state in relation to the megafauna dates on page 12 (paragraph 1): “*We included only direct AMS or conventional radiocarbon assays on hide, hoof, dung, and bone collagen; we excluded dates on apatite, whole bone, or bone mineral. We included genera that went extinct in North America as well as several well-defined extinct species that belong to genera that live on in the region, namely American lion, dire wolf, and Harrington’s mountain goat. So as not to oversample megafauna during the period they overlap with people, we followed Surovell et al.¹⁴ and excluded dates derived from unequivocal archaeological contexts. To this end, we followed the vetting of Grayson and Meltzer³² and excluded megafaunal dates from sites that represent clear kill and/or scavenging associations.*”

And in reference to the CARD dates we indicated on page 12 (paragraph 2): “*Given that we made use of minimally vetted and user-supplied radiocarbon data from the CARD, it was necessary to ensure data quality prior to our analysis. We followed the approach of Weitzel and Coddling⁶⁵ in removing all dates which are 1) duplicated in the database (based on the laboratory number), 2) marked as ‘anomalous’ by the uploader, and 3) reported to originate from paleobiological or geological (i.e., non-anthropogenic) contexts. Additionally, we removed radiocarbon dates (n = 54) derived from megafauna skeletal material that originated from sites where convincing kill/scavenging associations could not be made³² and thus would not relate to past human activity.*”

Reviewer #2 (Remarks to the Author):

This is an excellent paper, addressing a topic of wide interest with a clear set of hypotheses and an appropriate testing methodology. It is extremely clearly written and the methods used are very well explained and justified. The discussion is relevant and illuminating. I was initially concerned that they did not appear to be addressing Surovell et al's proposed correction for using summed radiocarbon probabilities as demographic proxies but they do get it to in the end and show in the SI that it does not make any difference to their conclusions.

Response: We appreciate Reviewer 2’s strong support of this manuscript.

Reviewer #3 (Remarks to the Author):

This paper is greatly to be praised for its hypothesis-testing approach, its very careful treatment of data, its innovative methodology, and its balanced, thoughtful discussion. In this it stands in very favourable contrast to many recent papers on late Quaternary extinctions, which are based on questionable modelling of poor-quality data (and mostly seeking to promote an overkill model of extinction, whereas I find the present contribution admirably even-handed). I do, however, have some significant comments of a methodological nature that I hope the authors will be able to address. To make the review process easier for myself and the authors, I have added line numbers to the text in the attached version.

I have three major concerns and various more minor ones:

(1) The first major concern is about the a priori restriction of possible climate-induced causality to the Younger Dryas (YD). The whole paper makes this assumption and it is stated explicitly: "Climate-based causes would thus be suggested in cases where megafaunal population declines that culminate in extinction occur during the YD...". What is the justification for this? What about the LGM, the late-glacial interstadial (GI-1) or indeed the warming of the early Holocene, as well as the YD (GS-1)? Why might not these have impacted megafauna too? A priori there is no reason why the YD should be singled out as the only chronological correlation with megafaunal decline that will be accepted as causal. Especially since your analysis looks at the whole 20-10 ka period, it is simply not necessary to set this up as such a limited hypothesis in advance. Now, in the event, the authors' data do show the YD as correlated with decline in many species, and that's fine, so the YD as a climatic cause can and should be discussed *after* we have seen those results, but not as an a priori assumption.

Response: We describe in some detail on page 5 (paragraph 2) the justification for focusing the climatic cause for extinctions on the YD. As we stated in the text: "*Climatic hypotheses for extinctions are varied but most relate to how changes in temperature, precipitation, and insolation seasonality resulted in habitat loss, simplification or fragmentation of plant communities, changes in plant nutrient quality and seasonal availability, or physiological stress^{1, 4, 6, 39-40}. Megafaunal population declines are thus anticipated to chronologically align with periods where these features converge. During the terminal Pleistocene, such a convergence becomes most apparent during the YD.*" In our view, framing the analysis this way provides the most reasonable justification to formulate a chronological expectation for the climatic hypothesis for Pleistocene extinctions. As the reviewers noted, the deductive, hypothesis-testing nature of the study is one of its virtues and we are reluctant to explore the climatic hypothesis inductively, without a priori expectations. Our decision, in this case is to leave the section as is but to add a final caveat at the end of the **Climate Change** section on page 5 (paragraph 2). That added sentence reads: "*Finally, climatic causes would also be suggested insofar as megafaunal and human populations are uncorrelated*

and extinctions converge during other chronological periods characterized by conditions unfavorable to large herbivores (e.g., Last Glacial Maximum, early Holocene)."

(2) Analogously to the above, why are only the Clovis people tested as possible overkill perpetrators? The date span goes to 10 ka, well beyond Clovis, but while Clovis is mentioned multiple times in the text there is nothing about the people in the post-Clovis interval. Even in Figure 2 only Clovis is indicated and any faunal changes after that time are attributed to climate. Yet the authors state that human populations over 5 millennia 'shared broadly similar technologies and foraging adaptations' (lines 315-318). In the first place, this statement needs further explanation and references. What cultures preceded and followed Clovis and how do you know they had similar adaptations etc? And if they had similar foraging adaptations, why might not they be implicated in the megafaunal declines & extinctions? This issue needs at the very least to be discussed.

Response: Although we emphasized that the overkill literature has focused on the role of Clovis hunters in megafaunal extinctions, we were (and are) explicit about the potential role of post-Clovis hunters as well. As we noted on page 5 in the final sentence of the Clovis Hunting section: "*Gradual or protracted overkill would be suggested by such correlations where significant megafaunal population declines leading to extinction occur after Clovis times.*" We also afford analytically a specific role for "*Post-Clovis Hunting*" in the causal summary in Table 1—where it is explicitly listed as a potential cause. As noted in Table 1 (footnote 2): "*[Extinctions are...] ... Consistent with post-Clovis hunting if declines to extinction begin after both Clovis times and the YD and significant negative relationships occur between humans and megafauna.*" We conclude in the current version that mammoth extinction in the Southwest is consistent with a multi-causal hypothesis where both YD climate change and post-Clovis hunting are implicated. To make it more clear that we explicitly examine the impacts of post-Clovis hunters on megafaunal extinctions in this paper we have changed the section heading on page 4 from "**Clovis Hunting**" to "**Human Hunting.**"

(3) I found the conclusions less believable than they otherwise would have been because of the imposition of an exponential growth curve as the null hypothesis for megafaunal population size. Animal populations do not go on growing exponentially forever; the general model in population biology is a logistic, S-shaped curve in which initial exponential increase then levels off. If we look at the sloth in Fig. 2 for example, in the interval 15-13 ka it develops a very steep exponential increase –it seems wholly unrealistic to project that through the roof as the null model. Instead, a logistic curve would form a more appropriate and believable null for the authors' study. At the very least it should be added as sensitivity study. It is not as though most of these megafaunal species had just entered the continent as

humans had; nor that they were expanding into the huge area vacated by the Laurentide ice sheet (because the bulk of your data is south of that). These considerations outweigh Shennan's supposed 'benefits' of an exponential model (line 379). Nor do I accept the Surovell model that preservation increases (and taphonomic loss decreases) exponentially, or in any regular manner at all, through the Late Pleistocene (lines 388-9). The Late Pleistocene is a very short interval relative to geological history, and climate-driven palaeoenvironmental effects on preservation potential (e.g. vegetation cover, frozen water), that we know to be strongly oscillating, are likely to be wholly dominant over a simplistic inverse chronological effect. I in any case felt muddled by the authors' account (lines 392-4) – is the modelled exponential increase in dates due to population growth, taphonomic effects, or some unknown combination of the two?

Response: While we appreciate Reviewer 3's concerns about our null exponential model, we have decided not to employ a logistic null model for several reasons. First, imposing a logistic growth model as a null would require an unacceptably arbitrary decision about the carrying capacity of each region specific to each taxon. The logistic growth equation requires a parameter for carrying capacity (typically "K") which regulates the growth of the modeled population. We see no valid way to assign a carrying capacity to these data. Were we to use a logistic null model, we would be forced to assume that the peak of each SPD represents carrying capacity and we have no *a priori* reason to think that. Simply because this peak represents the maximum population size of the taxon prior to climatic or anthropogenic effects does not mean that it represents carrying capacity. We are therefore missing crucial data to fit a logistic model to these SPDs. Second, we fit our null models using generalized linear modeling: a flexible regression technique in which the model can take a linear, exponential, logistic, or other form depending on parameterization and maximum likelihood estimation. We specified exponential fits for the reasons outlined in the manuscript (and by the reviewer above), but also experimented with logistic fits. Even when the generalized linear models were parameterized to fit logistically (under the unfounded assumption discussed above that the SPD peak represents carrying capacity), they fit themselves exponentially resulting in no difference to our null models. The internal optimization algorithm of the models identifies these SPD data as behaving exponentially, not logistically, and therefore fits the null model exponentially. Third, a null model is just that: a baseline against which to compare data. It is not an assertion about what reality should look like or a prediction about megafauna population dynamics that is intended to be accurate. In this respect, we feel that an exponential null model serves its purpose of providing an expectation against which we can identify population declines in these data. For these reasons, we have elected to retain exponential null models in our analyses.

Minor comments by page/line number:

4/73-76: These conclusions of previous authors should be cited a little more critically. For example,

radiocarbon data from S America is still very sparse, and even for N Am many extinct taxa have few or no dates.

Response: We have added a more critical perspective of this work. It reads (page 4, paragraph 1): “*Much of this work, and all of it focused on North America south of glacial ice, has approached the extinctions issue by reconstructing trends in the megafauna population as a single entity which may obscure significant species level detail and variability.*”

5/106: ‘these features converge’ I found ambiguous. You mean ‘several of these factors coincide’?

Response: We changed converge to *coincide*.

5/110: it is unclear what is meant by ‘this interglacial cycle’ in the context of a discussion about the Younger Dryas (YD). The YD is not a glacial but a stadial. Or do the authors mean the whole last glacial/interglacial cycle?

Response: The comparison is between the onset of the current interglacial cycle (that begins near the YD) with previous earlier interglacials. We adjusted the relevant text to now read: “*...may have been unique to the transition into the current interglacial cycle relative to earlier ones*”.

5/112-116: having acknowledged that there is geographic variation in the palaeoenvironmental effects of the YD, the authors should state specifically what is known of these changes in N Am S of the ice (their study area). And not just for the YD but for the whole GI1-GS1-MIS1 cycle.

Response: We added a clause to the sentence below to highlight the difficulty of generalizing the paleoenvironmental conditions of the YD. But the critical conditions of the YD as they should affect large mammals, and of particular relevance here, are included in the second part of the sentence. “*Although it is difficult to generalize the conditions of the YD across North America—whether it was, for instance, cool/dry or warm/wet—as the climate of the period clearly varied spatially, temporally, and seasonally⁴⁴, many YD paleoclimatic records suggest an abrupt return to glacial-like average annual temperatures, combined with peaks in insolation seasonality (i.e., cold winters and hot summers) and rapidly growing atmospheric CO₂ concentrations⁴³⁻⁴⁶.*”

6/152-3: for consistency, give all species full binomen.

Response: Throughout the paper, we list the complete species binomial or genus names for each taxon at the first usage in the paper and thereafter use the accepted common names.

7/155: does 'postdating 20.0 ka' mean 20-10 ka or 20-15 ka?

Response: We changed the relevant text to now read: *"The sample dating between 20.0 and 10.0 ka (n = 332) is similar in composition..."*

7/166: is the dating quantity and precision good enough to demonstrate 'a brief return to exponential growth' (or any other pattern) in a period of only 100 yr (12.8-12.7 ka)?

Response: While interpreting much about small changes in these SPDs can be problematic, here we are simply reporting our model results. The methods employed here have been shown to accurately reflect patterns on a scale greater than 200 years (see citations for Shennan, Timpson, and colleagues) and the brief return to exponential growth described here is predicted to have lasted for 233 years. Caution is nonetheless advisable when interpreting smaller changes in the SPD, and for this reason we avoid making any such interpretations about causes of smaller fluctuations in the SPDs instead choosing to focus only on terminal declines and overall correlations between megafauna and human SPDs.

7/169-70: 'Significant negative correlations would be consistent with a human hunting impact...' – or just preferring different climates/habitats (the reciprocal of the inferred human-sloth relationship)?

Response: This is a valid point but it is subsumed under the overarching assumption of the general approach that summaries of date frequencies (SPDs) can be used as populations proxies. Further, we do not suggest that significant negative correlations must equate with human overkill, only that they are *consistent with* human caused extinctions. In attempt to further convey our understanding of this potential equifinality we have added on page 5 (paragraph 1), in the section where we develop this logic the phrase "*other things equal*". The full passage now reads: *In any case, we expect that if Clovis hunting drove extinctions, other things equal, megafaunal population declines should occur during Clovis times and be associated with significant human population increases. In other words, negative correlations should exist between human and megafaunal populations.*

7/173-175 (and more generally): For the whole continent, can you affirm that humans & megafauna were even in the same areas? If not, your tests for interaction would be invalid.

Response: We certainly cannot confirm a priori that people were in the same areas or encountered megafauna other than in the few (15) sites that show human hunters utilized megafauna, as described on page 9 (paragraph 2). The test as constructed is, however, still valid. If humans and megafauna were not in the same areas and humans were thus not hunting them, then we would not anticipate expanding human populations to have any influence on megafauna populations and thus we would expect no significant negative correlations between their populations.

9/221: For extinctions in northern Eurasia it seems odd not to cite any of the numerous papers by Stuart & Lister who have done the most work in that region.

Response: We added the following citation in this context:

59. Stuart, A.J., Lister, A. M. Extinction chronology of the woolly rhinoceros *Coelodonta antiquitatis* in the context of late Quaternary megafaunal extinctions in northern Eurasia. *Quaternary Science Reviews* 51, 142-143 (2012).

10/247: With no disrespect to Widga and colleagues, I was slightly bothered by the apparently uncritical acceptance of their extinction scenario that is supported by hardly any dating of the carnivore extinctions or this improbable sounding mechanism of extinction through overpopulation. You say your data are 'consistent' with it but in fact they provide no evidence for this specific scenario – I would prefer if you added that!

Response: We amended the relevant section as follows: *“Although robust tests of this specific scenario have yet to be conducted, the results from our Great Lakes analyses are not inconsistent with it insofar as we detect no human role in the extinction process in this area, the significant demographic busts that lead to extinction for mastodon and mammoth occur during the YD, and none of our collection of directly dated large carnivores (n = 134) are from specimens derived from the Great Lakes region (SI Table 1).”*

10/256: 'formerly' with respect to what/when?

Response: We omitted “formerly less abundant” from this sentence.

11/264: what is the relevance of bison being aggressive?

Response: It may be relevant to how they interacted with, and potentially outcompeted, other megafauna which is the main point of this section.

11/268-271. I found this passage unconvincing, resting on a string of doubtful assumptions. How is it known that foraging efficiency decreased? By exclusion? Is the change in age structure of the fossil assemblage real or taphonomic (or collecting bias)? And the idea that cold selects for large body size (Bergmann's Rule) is much debated.

Response: The suggestion here was not intended to be indisputable. The argument is that bison size declines during the YD (assuming that the trend is real and not a taphonomic artifact; we cite Hill et al. 2009 for details on the trend) and that the decline is absent any apparent increase in human harvest pressure (that should be reflected by declines in mean/maximum age—that is not apparent). Conventional arguments suggest, other things equal, that cold (YD) climate should select for larger size, but the pattern is in the opposite direction. This leaves depressed juvenile nutritional quality/quantity (foraging efficiency) as a viable cause for the decline in size. We conveyed a more suggestive tone to this argument by changing the beginning of the sentence that originally read “also appears linked to declines in foraging efficiency” to “*may also be linked to declines in foraging efficiency*”.

11/278-9: As I am sure the authors realise, taxa may also lack a radiocarbon record because researchers tend to expend resources building chronologies of species that already have dates... or those species might have failed dating because all the bones are too old.

Response: Yes, this is a potential bias when using SPDs as a proxy for population size. We hope to have subsumed this specific concern as well as others in the introductory clause of the sentence that states: “*Insofar as relative date frequencies track population sizes...*”

12/298: maybe ‘arbitrary’ is not quite the right word; I imagine it was carefully chosen as the most broadly acceptable earliest likely date for colonization. As an aside, the arguments over the date of ‘initial colonization’ can perhaps be placed to one side by focusing on the earliest significant colonization of the continent (and hence plausible impactors on megafauna).

Response: We edited the relevant passage to now read: *...between 15.0 and 10.0, the latter chosen as the most widely accepted earliest date for the initial colonization of the Americas.*

12-13/315-318: As well as the issue about Clovis discussed above, I found this passage hard to understand. (1) “variation in energy consumption” – do you mean per capita energy consumption? (2) “a factor that can influence the relationship between carbon production and population levels” – I am

afraid I couldn't understand what this meant. And (3) "can be held constant in this case" – please explain why is this significant?

Response: We clarified this passage to now read: *As it has been argued that variation in economic complexity can influence per capita energy consumption and the production of dateable archaeological materials (bone, charred wood, etc.), semi-independent of population size⁶⁹, we observe that our archaeological date samples were derived from human populations that shared broadly similar technologies, foraging adaptations, and levels of economic complexity.*

13/334: recommend keeping dates in chronological order, so 15.0-11.0 ka.

Response: We corrected as suggested.

16/403-4: The authors are to be praised for their careful approach to date auditing. But various other authors have discussed these issues so it would not be inappropriate to cite them – e.g. in relation to the point discussed here, Lister & Stuart (2013) applied a 1980 cut-off to exclude dates made before proper washing of collagen became routine.

Response: We elaborated on our approach to date auditing in response to the comments of Reviewer 2 on their notes pertaining to page 16, paragraph 2, in the original manuscript (see above) and now found in the first and second paragraphs of the **Methods** section. Several citations to previous approaches to date auditing are provided here.

There are also a very few minor suggested textual alterations in the attached version.

Reviewers' Comments:

Reviewer #1:

Remarks to the Author:

This is my second review of this manuscript. This is an excellent study examining the population dynamics of humans and megafauna during the late Pleistocene and should be of interest to a wide variety of disciplines. Nevertheless, in my first review, I argued for key changes to the presentation and the supporting analysis. Although I made a series of suggestions related to minor, text-specific statements, my primary criticism was of the authors' decision to group the majority of megafauna used in their analysis into a "all other" category. As I noted earlier, this decision obscured a huge amount of detail that was potentially important in understanding variation. I also suggested that authors might want to compare their results from extinct species to megafauna that survived the Pleistocene extinction, such as Bison, Ovis, Odoceileus, and Alces.

I was very pleased with the revised manuscript, including supplemental information and the detailed response letter. The changes to this manuscript adequately address all my minor and major concerns. The authors' decision to model separate SPD for mammoth, mastodon, ground sloth, horse and saber-tooth cat reflect a significant and well-reasoned change to the manuscript, in line with my previous recommendations. In addition, the addition of an SPD analysis of Bison adds to the persuasive nature of the authors' argument, and satisfies my other major suggestion.

There are still several minor issues in the current version of the text, both of which should be easy to correct:

- On page 4 there is an inserted statement in the margin of the text concerning the location of prior studies that have considered human overkill. The comment, apparently written by Elic Weitzel, seems to identify a mistake in the text. The authors should double check that this statement in the text is correct.
- On page 11 (Discussion Section), the author provide the results of an SPD for bison. As stated above, this is an excellent addition to the paper, but I believe there is no discussion of the methods or original dataset used in this analysis in either the Modeling Population (pp. 6-8) or the Methods (pp. 12-16) sections of the paper. The original R code presented in the supplementary material does indicate that the bison dates were compiled from CARD, Heintzman et al. 2016 and MacDonald 1981. I suggest the authors include a more explicit statement concerning this analysis in the main body of the text.

Overall, this is a much-improved manuscript that makes a substantial contribution to the field of study; as a result, I recommend that the editor of Nature Communications should accept this paper.

Reviewer #3:

Remarks to the Author:

(1) I'm afraid I still don't agree with the authors that the Younger Dryas is a priori uniquely qualified to act as the sole or leading hypothesis for climatic causation of extinctions. For many authors the rapid warming (more rapid than the onset of the YD) and afforestation of the Bolling/Allerod interstadial has been considered at least as potentially significant (e.g. Stuart & Lister 2012, Cooper et al. 2015). I am prepared to respect the authors' wish to emphasise the YD as their lead hypothesis, but instead of a single sentence right at the end to passingly acknowledge other potentially causal climatic episodes (and currently not even mentioning the B/A), a balanced account of all the potential climatic drivers from LGM to Holocene, should be given at the start of the discussion, then leading into an explanation of why the YD is singled out.

(2) Explanation largely accepted, but I did not (and would not now, except that the authors told me to) read the sentence "Gradual or protracted overkill would be suggested by such correlations where significant megafaunal population declines leading to extinction occur after Clovis times" to be a clear statement that subsequent cultures might have been responsible. Please make it more explicit.

(3) I remain doubtful that the exponential model is appropriate as an 'expectation against which we can identify population declines'. That is precisely the problem: if your model is a positive exponential increase (which it is for all the species in Fig. 2, some of them quite steeply so), then you are almost guaranteeing that reality will show a 'decline'. I accept the authors' point about having no basis for parameterising a logistic curve, but an exponential increase into an unknown future is equally speculative. The fact that their model fits an exponential curve up to the chronological cut-off does not in itself give us any reason to expect the population would (even without human or climatic impact) continue along the same trajectory after it. All I asked for was some sensitivity testing. A horizontal straight line (i.e. assuming the population just stays the same after the cut-off) would actually be the most neutral (i.e. least theory-laden) model against which to test reality.

Dear Editor,

Please find our point-by-point response to the reviewer's comments below. These changes are also highlighted in the main text and Supplementary Information sections.

Jack M. Broughton and Elic M. Weitzel

Reviewer #1 (Remarks to the Author):

This is my second review of this manuscript. This is an excellent study examining the population dynamics of humans and megafauna during the late Pleistocene and should be of interest to a wide variety of disciplines. Nevertheless, in my first review, I argued for key changes to the presentation and the supporting analysis. Although I made a series of suggestions related to minor, text-specific statements, my primary criticism was of the authors' decision to group the majority of megafauna used in their analysis into a "all other" category. As I noted earlier, this decision obscured a huge amount of detail that was potentially important in understanding variation. I also suggested that authors might want to compare their results from extinct species to megafauna that survived the Pleistocene extinction, such as Bison, Ovis, Odoceileus, and Alces.

I was very pleased with the revised manuscript, including supplemental information and the detailed response letter. The changes to this manuscript adequately address all my minor and major concerns. The authors' decision to model separate SPD for mammoth, mastodon, ground sloth, horse and saber-tooth cat reflect a significant and well-reasoned change to the manuscript, in line with my previous recommendations. In addition, the addition of an SPD analysis of Bison adds to the persuasive nature of the authors' argument, and satisfies my other major suggestion.

There are still several minor issues in the current version of the text, both of which should be easy to correct:

- On page 4 there is an inserted statement in the margin of the text concerning the location of prior studies that have considered human overkill. The comment, apparently written by Elic Weitzel, seems to identify a mistake in the text. The authors should double check that this statement in the text is correct.
- On page 11 (Discussion Section), the author provide the results of an SPD for bison. As stated above, this is an excellent addition to the paper, but I believe there is no discussion of the methods or original dataset used in this analysis in either the Modeling Population (pp. 6-8) or the Methods (pp. 12-16) sections of the paper. The original R code presented in the supplementary material does indicate that the bison dates were compiled from CARD, Heintzman et al. 2016 and MacDonald 1981. I suggest the authors include a more explicit statement concerning this analysis in the main body of the text. Overall,

this is a much-improved manuscript that makes a substantial contribution to the field of study; as a result, I recommend that the editor of Nature Communications should accept this paper.

Response: We corrected the marginal note on page 4 and provided a discussion of the Bison SPD analysis in the methods section on page 14. The discussion is as follows: *“As noted above, our radiocarbon datasets for extinct megafauna are based exclusively on directly dated specimens derived from paleontological contexts following the systematic evaluation of Meltzer and Grayson³². Since no such evaluation has been provided for the Bison radiocarbon record, our analysis of this taxon (Fig. 6) is necessarily based on the collective assemblage of dates derived from both archaeological and paleontological contexts. Since the radiocarbon dates for Bison are to some degree a subset of the broader human radiocarbon record, a bison-human correlation analysis would be inappropriate and, thus, was not conducted. The sample of direct dates for Bison (n = 89) was compiled from the CARD⁵⁸, Shapiro et al.¹⁹, and McDonald⁶⁰.”*

Reviewer #3 (Remarks to the Author):

(1) I'm afraid I still don't agree with the authors that the Younger Dryas is a priori uniquely qualified to act as the sole or leading hypothesis for climatic causation of extinctions. For many authors the rapid warming (more rapid than the onset of the YD) and afforestation of the Bolling/Allerod interstadial has been considered at least as potentially significant (e.g. Stuart & Lister 2012, Cooper et al. 2015). I am prepared to respect the authors' wish to emphasise the YD as their lead hypothesis, but instead of a single sentence right at the end to passingly acknowledge other potentially causal climatic episodes (and currently not even mentioning the B/A), a balanced account of all the potential climatic drivers from LGM to Holocene, should be given at the start of the discussion, then leading into an explanation of why the YD is singled out.

Response: We included much more discussion indicating our openness to the possibility that other climatic periods may have potentially contributed to extinctions.

We added the following discussion (2nd paragraph, page 5) to address this concern:

“Megafaunal extinctions resulting from these processes have been suggested for a range of different late Pleistocene climatic episodes but recent work has suggested strong associations between extinction events and warm interstadials associated with Dansgaard-Oeschger events^{2,3,9}, such as the Bølling-Allerød (B-A; ~14.7 to 12.9 ka), as well as the unique conditions of the YD, as noted above. For instance, using ancient DNA and radiocarbon records, Cooper et al.² derive detailed time series for multiple megafauna genetic clades or species in both Eurasia and Eastern Beringia and show that the period encompassing the B-A and YD is characterized by a higher frequency of regionwide extirpations or global extinctions than any other over the last 25,000 years. Since it is

known that many now extinct genera of North American megafauna survived the B-A^{1,12,13,14}, attention has focused on the potential role of conditions during the YD in driving extinctions in this region^{5,12,15,16}.”

We also included reference to the B-A in the last paragraph of the climate section on page 6:

“Climate-based causes would thus be suggested in cases where megafaunal population declines that culminate in extinction occur during the B-A or YD and where human and megafaunal populations are positively correlated—rising and falling together—or uncorrelated.”

And in the first paragraph of the **Discussion** we added: *“Since all taxa exhibit population growth consistent with the null models through the pre-Clovis B-A with busts leading to extinction occurring either during Clovis times or the YD, we focus on the period covered by the latter phenomena in addressing the causes for extinctions.”*

(2) Explanation largely accepted, but I did not (and would not now, except that the authors told me to) read the sentence “Gradual or protracted overkill would be suggested by such correlations where significant megafaunal population declines leading to extinction occur after Clovis times” to be a clear statement that subsequent cultures might have been responsible. Please make it more explicit.

Response: We added the following sentence at the end of the first paragraph, page 5, to make this point more explicit: *“Overkill resulting from subsequent cultures (e.g., Folsom) would be suggested by such negative correlations where significant megafaunal population declines leading to extinction occur after Clovis times.”*

(3) I remain doubtful that the exponential model is appropriate as an ‘expectation against which we can identify population declines’. That is precisely the problem: if your model is a positive exponential increase (which it is for all the species in Fig. 2, some of them quite steeply so), then you are almost guaranteeing that reality will show a ‘decline’. I accept the authors’ point about having no basis for parameterising a logistic, but an exponential increase into an unknown future is equally speculative. The fact that their model fits an exponential curve up to the chronological cut-off does not in itself give us any reason to expect the population would (even without human or climatic impact) continue along the same trajectory after it. All I asked for was some sensitivity testing. A horizontal straight line (i.e. assuming the population just stays the same after the cut-off) would actually be the most neutral (i.e. least theory-laden) model against which to test reality.

Response: We have now directly addressed this issue by providing further discussion in the manuscript regarding our concerns with using logistic null models, but also include a new section in the Supplementary Information where we present all analyses using logistic rather than exponential null models. After experimenting with several methods for fitting logistic models to these SPDs (most of which failed to fit identifiable models), we ended up using Richard’s Equation for a five-parameter logistic growth model. Fitting such a model worked in cases where

fewer-parameter models failed as the SPDs largely behave exponentially. Maximum likelihood estimation of optimal parameter values also allowed us to avoid the issue of specifying carrying capacity *a priori*. This approach led to logistic models which often fit these SPDs as if terminal Pleistocene increases were the lower portion of logistic curves before the inflection point is reached and slope declines. Some SPDs better fit logistic models, but any such slight improvements in model fit came at the expense of several additional degrees of freedom.

The following provides the added text that addresses this issue:

End of 1st paragraph, page 7: *“The null models to which we compare each SPD represent the exponential growth trajectories that the human and megafaunal taxa followed prior to observed declines, however we also fit logistic null models to explore the potential effect of null model choice on our results (see Methods).”*

And in the 1st paragraph of the subheading *The Utility of Comparisons to an Exponential Null Model*, page 16-17 we added:

Of course, real-world populations are rarely unregulated, making a logistic growth model a generally more realistic representation of population dynamics. However, applying a logistic growth model to these data present challenges. Logistic population growth models reach their top asymptote due to the effect of carrying capacity, which constrains a population from growing beyond a certain point typically dictated by food supply. It is not possible to derive the terminal Pleistocene carrying capacity for each of these megafauna taxa, making it problematic to assign such values in constructions of null models. We do not know the available food supply for each taxon, nor should we assume that each taxon had reached its carrying capacity prior to its declines towards extinction. Nonetheless, to ensure that our choice of exponential null models does not alter our results in any substantial way, we repeated all analyses using logistic null models (Supplementary Figs. S6 – S8). These models were fit using maximum likelihood estimation of five-parameter logistic models based on Richard’s Equation. A maximum likelihood approach permits the estimation of the carrying capacity parameter (and all other parameters) based on the characteristics of the observed data, not any a priori expectation. Such values may not reflect real-world carrying capacity, for example, but are an empirically-robust way of fitting such a model to these data in the absence of other information. Attempts to fit the more commonly known three-parameter logistic model typically failed as most of these SPDs behave exponentially prior to their breakpoint dates, not logistically. Many of the logistic models fit here are therefore either poor fits to these data or obtain marginally better fits at the cost of several additional degrees of freedom. For this reason, these null models frequently resemble their exponential

counterparts because maximum likelihood estimation places the top asymptote (i.e. carrying capacity) far above the SPD in order to optimize the model fit to the available data.

And beginning on Page 2 of the Supplementary Information section we added the following new section:

“Supplementary Information: Analysis Using Logistic Null Models

Supplementary Figs. S6-S8 show the SPDs for the human and various megafaunal taxa and statistically significant deviations using logistic, rather than exponential, null models. The results shown here for all regions and taxa are in almost all cases very similar to the analyses conducted with exponential null models with notable differences occurring only with US mammoth, US mastodon, and US Equus. In the case of US mammoth, a brief return to trend occurs during the early YD after a brief bust toward the end of the Clovis period. The terminal decline for mammoth thus occurs during the YD rather than in Clovis times as indicated in the analysis based on an exponential model. The decline during the Clovis period is still evident, however, and given the significant correlation with the human SPD is in any case consistent with a human role in mammoth extinction. For US mastodon, the terminal decline date shifts into the Clovis period from its place in the YD with the exponential null model. However, since the mastodon and human SPDs do not exhibit a meaningful correlation, the difference in this case pertains only to the timing of the apparent climatic influences on this taxon—effectively pushing the terminal decline back ~400 years. Finally, for US Equus, there are several pre-Clovis “busts” while the overall trend of the SPD is still increasing. This is due to a poorly-fit logistic null model with a top asymptote far above the highest SPD value. When simulating the logistic null model confidence interval, this has the effect of dragging the interval upwards for this portion of the curve, resulting in the discrepancy observed here. However, this result is due to a poorly fit model and should be treated with skepticism: it is quite clear from visual inspection of this SPD that such “busts” occur during a period of general increase and are therefore not related to declines towards extinction. In sum, the overall trends and general conclusions regarding the causative factors relating to megafaunal extinctions are robust to the use of either exponential or logistic null models.”

Reviewers' Comments:

Reviewer #3:

Remarks to the Author:

I am now very satisfied that my concerns have been addressed. In particular, the authors have provided a clear context and explanation for their focus on the Younger Dryas, as well as extra analysis leading to extended discussion of the issues around the choice of a null model.